# Mechanisms of life cycle simplification in African trypanosomes

Guy R. Oldrieve [1] ✉, Frank Venter [1], Mathieu Cayla [2], Mylène Verney[3], Laurent Hébert [3], Manon Geerts[4,5], Nick Van Reet[5] & Keith R. Matthews [1] ✉

African trypanosomes are important parasites in sub-Saharan Africa that undergo a quorum-sensing dependent development to morphologically 'stumpy forms' in mammalian hosts to favour transmission by tsetse flies. However, some trypanosome clades have simplified their lifecycle by escaping dependence on tsetse allowing an expanded geographic range, with direct transmission between hosts achieved via blood-feeding biting flies and vampire bats (*Trypanosoma brucei evansi*, causing 'surra') or through sexual transmission (*Trypanosoma brucei equiperdum*, causing 'dourine'). Concomitantly, stumpy formation is reduced and the isolates are described as monomorphic, with infections spread widely in Africa, Asia, South America and parts of Europe. Here, using genomic analysis of distinct field isolates, we identify molecular changes that accompany the loss of the stumpy formation in monomorphic clades. Using CRISPR-mediated allelic replacement, mutations in two exemplar genes (Tb927.2.4020; Tb927.5.2580) are confirmed to reduce stumpy formation whereas another (Tb927.11.3400) is implicated in altered motility. Using laboratory selection we identify downregulation of RNA regulators as important in the initial development of monomorphism. This identifies a trajectory of events that simplify the life cycle in emergent and established monomorphic trypanosomes, with impact on disease spread, vector control strategies, geographical range and virulence.

T*rypanosoma brucei* subspecies are the causative agents of human African trypanosomiasis (HAT) and Animal African trypanosomiasis (AAT). These parasites have a dixenous life cycle, entailing transmission between the mammalian host and the tsetse fly vector, this complex life history having evolved independently multiple times from monoxenous trypanosomatid ancestors[1]. Dixenous *T. brucei* are pleomorphic, involving a developmental transition from a proliferative 'long slender' form to a cell-cycle arrested 'short stumpy' form, which is preadapted for uptake by the tsetse fly[2,3]. The transition from slender to stumpy forms involves a density-dependent process where parasite-

released peptidases hydrolyse extracellular proteins to produce oligopeptides as a quorum sensing signal[4–6]. These are received via TbGPR89, and initiate a signalling cascade which induces developmental progression[7,8] in a process that can be recapitulated in vitro using oligopeptide-rich broth[9].

The complex lifecycle of *T. brucei* restricts these parasites to the geographic range of the tsetse fly within Africa. However, some clades have simplified their life cycle by foregoing development in their arthropod vector, allowing them to escape the tsetse belt, causing disease in Asia, South America, and Europe[10]. These have been

[1]Institute for Immunology and Infection Research, School of Biological Sciences, University of Edinburgh, Edinburgh EH9 3FL, UK. [2]York Biomedical Research Institute and Department of Biology, University of York, York, UK. [3]Unité Physiopathologie et Epidémiologie des Maladies Equines (PhEED), Laboratoire de Santé Animale, Site de Normandie, Agence Nationale de Sécurité Sanitaire de l'Alimentation, de l'Environnement et du Travail (ANSES), 1180 route de l'église, 14430 Goustranville, France. [4]Department of Biology, Katholieke Universiteit Leuven, Leuven, Belgium. [5]Department of Biomedical Sciences, Institute of Tropical Medicine, Antwerp, Belgium. ✉e-mail: guy.oldrieve@ed.ac.uk; keith.matthews@ed.ac.uk

historically described as separate species, namely *T. evansi* and *T. equiperdum*[11], defined by their morphology, host species and disease presentation. *T. evansi* was described as the causative agent of the disease surra, exploiting mechanical transmission via blood feeding biting flies (tabanid and stable flies), whereas *T. equiperdum* was described as the causative agent of dourine, being sexually transmitted between equids[12]. Without tsetse transmission, each is characterised by their reduced production of arrested stumpy forms and so are described as monomorphic. Having dispensed with the tsetse fly vector and the associated metabolic needs at that stage of the life cycle, monomorphic *T. brucei* frequently exhibit a reduced or absent mitochondrial genome, the kinetoplast (kDNA), as a secondary consequence[13]. Since *T. brucei* sexual reproduction occurs in the tsetse fly salivary gland[14,15], a further consequence of the simplified life cycle of monomorphs is that they are obligately asexual, the parasites being constitutively diploid in their mammalian host. Although this transition facilitates the expansion of monomorphic *T. brucei* into new ecological niches, in the long term obligate asexual reproduction can limit the ability of an organism to purge deleterious mutations whilst also restricting the maintenance of population-wide genetic diversity, hindering adaptation[16,17]. Recent analyses revealed monomorphic *T. brucei* clades are polyphyletic, with each clade separated by pleomorphic isolates[18–21]. These are described as 'types'; *T. b. equiperdum* type BoTat, *T. b. equiperdum* type OVI, *T. b. evansi* type A and *T. b. evansi* type B[20,21], although an alternative nomenclature based on 'ecotypes' has also been proposed[22]. Whilst mechanisms to adapt to kDNA loss have been described[23], the molecular events which originate a simplified lifecycle are unknown.

Here we analyse the genomic sequences of a large number of pleomorphic and monomorphic parasite isolates to identify genes with the potential to contribute to life-cycle simplification. Several monomorphic gene sequences, that contain clade specific mutations, confer reduced developmental competence after allelic replacement in a plemophic line. Furthermore, gene expression changes that occur as parasites are selected in the laboratory for reduced quorum sensing highlight a common profile of changes associated with RNA regulation. In combination this provides insight into how trypanosomes can reduce dependence upon tsetse flies for their transmission.

## Results

We reasoned that the occurrence of multiple naturally occurring monomorphic clades, alongside the ability to generate new monomorphic lines by selection against stumpy formation in vitro, could provide insight into the mechanisms underlying simplification of the parasite lifecycle and the evolution of monomorphism. Therefore, 83 *T. brucei* isolates from diverse hosts such as humans, cattle, equids, camel and tsetse (Supplementary Data 1), including 37 monomorphic isolates, were analysed for their phylogenetic relationships. The resulting phylogeny corroborates earlier evidence for at least 4 independent origins of monomorphic *T. brucei* (Fig. 1a)[19–21]. As previously noted, the isolate IVM-t1, which was isolated from the genital mucosa of a horse and typed as *T. equiperdum*[24], groups with *T. b. evansi* type B isolates but shares a relatively distant relationship to other isolates designated in that clade, suggesting that IVM-t1 could represent a 5th independent monomorphic clade[21] (Fig. 1a).

### Independent monomorphic clades display unique hallmarks of transition to an obligate asexual lifecycle

At the chromosome level, we did not observe enrichment in the normalised density of mutations between clades, however, peaks and troughs of mutation density are found in smaller regions, such as *T. b. evansi* type A on chromosome 8 (Fig. 1b−SNP track). Distinct from *Leishmania* spp. where remarkable copy number variation (CNV) plasticity provides a mechanism to rapidly respond to environmental stimuli[25], we did not identify clade specific CNV in monomorphic *T.*

*brucei* at the chromosome level (Fig. 1b−CNV track), corroborating previous CNV studies in *T. brucei* subspecies[26]. However, *T. b. equiperdum* type OVI does display smaller CNV, the largest being a duplicated region of Chromosome 7. This region spans 46 genes, which are significantly enriched for gene ontology (GO) molecular functions; 'glutathione peroxidase activity', 'peroxidase activity', 'oxidoreductase activity acting on peroxide as acceptor' and 'antioxidant activity' (Fig. S1a) as the region contains the three TDPXs (tryparedoxin-dependent peroxidases) which are part of the antioxidant defence system and localise to the mitochondrion and cytosol[27].

Being obligately asexual and so unable to undergo sexual recombination, we sought to identify evidence of the accumulation of mutations in monomorphic clades by calculating the diversity of nonsynonymous to synonymous mutations (dN/dS). Although we did not find broad enrichment of genes with an accumulation of deleterious mutations at the chromosome level (Fig. 1b−dN/dS track), each clade displayed >1000 genes which had a positive selection (dN/dS >1), highlighting an accumulation of nonsynonymous mutations in a monomorphic clade but neutral (dN/dS = 1) or purifying (dN/dS <1) selection in the pleomorphic background (Supplementary Data 3). Mitochondrial GO components were common amongst these sets of genes for all clades aside from *T. b. evansi* type B (Fig. S1b−f), likely reflecting reduced dependence upon mitochondrial processes with the exclusion of tsetse stages in monomorphic clades. *T. b. evansi* type B GO enrichment for mitochondrial membrane components fell below our significance threshold but was significantly enriched for membrane components (Fig. S1c). Only 55 genes (Supplementary Data 2) were under positive selection in all monomorphic clades, but these were not significantly enriched for GO components, broadly suggesting that monomorphism is associated with an accumulation of mutations in mitochondrial components, but that specific components are unique to each clade. Focusing on groups of known genes associated with the QS pathway or those with enriched expression in the insect stage of the parasite's life cycle, QS genes have an increased dN/dS ratio in *T. b. equiperdum* type OVI and *T. b. equiperdum* type BoTat compared to essential genes (Fig. S1g).

Finally, we examined runs of homozygosity (ROH) which can highlight regions of the genome that have undergone gene conversion as a mechanism for asexual organisms to rid themselves of deleterious mutations in the absence of sexual recombination[17]. This identified clade specific ROH on chromosomes 10 and 11 in *T. b. equiperdum* type OVI and *T. b. equiperdum* BoTat, respectively (Fig. 1b−ROH track). The ROH in *T. b. equiperdum* type BoTat covers 1,004 genes but is not enriched for a particular GO term, whereas the 875 genes covered by the *T. b. equiperdum* type OVI ROH was enriched for GO components 'transcriptionally silent chromatin', 'replication fork', and 'non-membrane bound organelles' (Fig. S1h).

### Identification of molecular changes that reduce stumpy formation

To identify changes likely to contribute to the life cycle simplification, clade-specific mutations were experimentally targeted. These were selected based on (i) their presence in a monomorphic clade but not in pleomorphic isolates, (ii) association with QS, and/or (iii) with a positive dN/dS ratio. Of these, an exemplar subset of genes were functionally characterised via a pipeline involving locus-specific replacement of both alleles in developmentally competent *T. brucei* AnTat1.1 using CRISPR-Cas9, and add-back replacement, with each step mirrored in parallel using wild-type allelic replacements to control for transfection and culture-associated loss of pleomorphism. For each cell line, developmental capacity was assessed using growth in vitro in the presence or absence of brain-heart infusion (BHI) broth as a source of QS oligopeptides[4,28], with cell cycle arrest in G1/G0 and expression of the stumpy-specific marker protein PAD1[29] monitored at the individual cell level. In combination these define

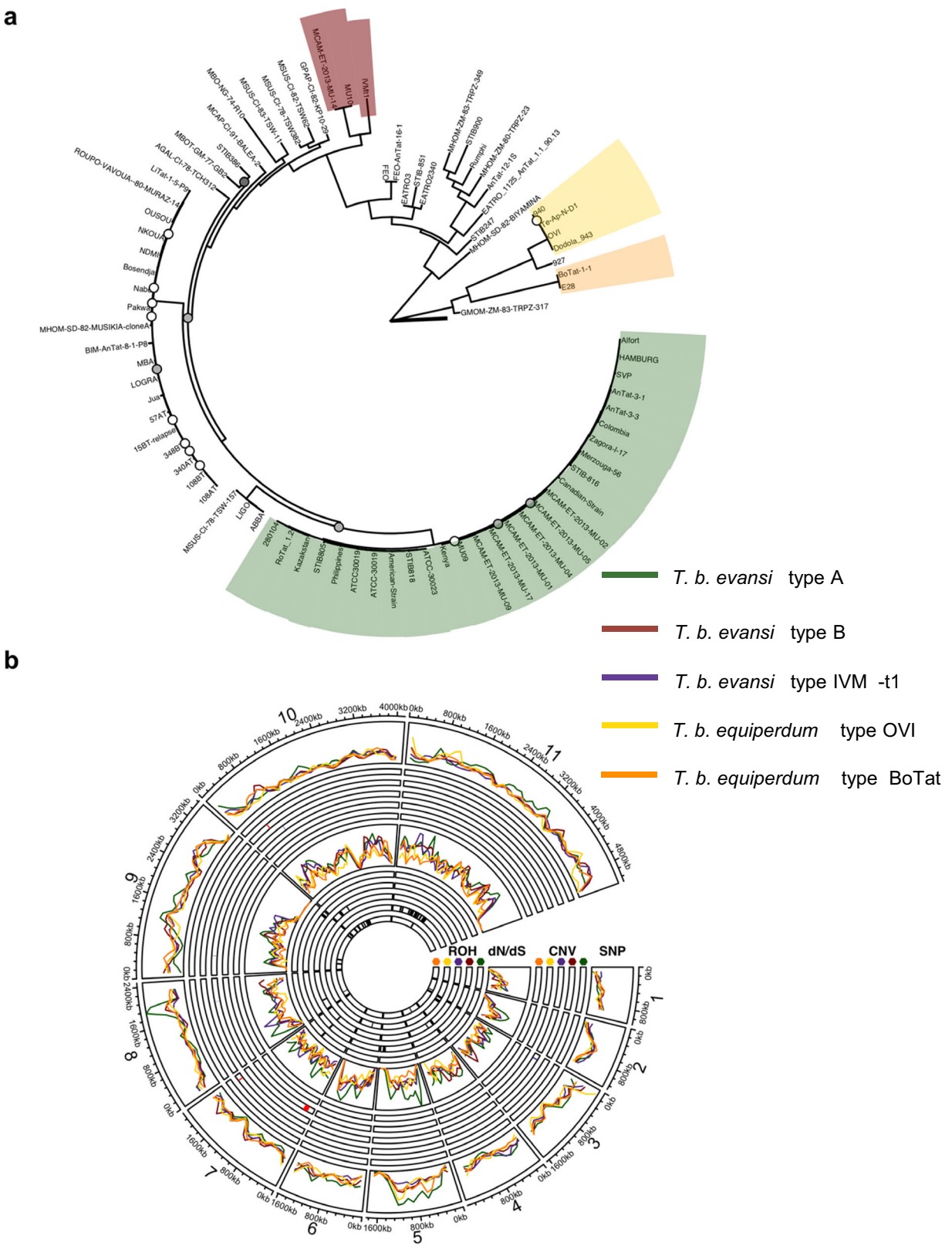

**Fig. 1 | Monomorphic *T. brucei* are polyphyletic and form at least four independent clades, each displaying clade specific features at the genome scale.**
**a** A centrally rooted phylogenetic tree was created with 368,730 SNPs. Bootstrap confidence below 100 are reported by circles; white: 75–100, grey: 50–75 and black: 25–50. The tree scale, highlighted by the bold line at the tree root, represents 0.05 substitutions per site. Monomorphic isolates form at least four distinct clades, *T. b. equiperdum* type OVI, *T. b. equiperdum* type BoTat, *T. b. evansi* type A and *T. b. evansi* type B. IVM-t1, an isolate originating from the genital mucosa of a horse, groups with *T. b. evansi* type B. Pleomorphic clades are not highlighted. A full list of the isolates used in this tree is found in Supplementary Data 1. **b** The density of clade-specific mutations, location of copy number variation (CNV) highlighted in red (increase in copy number) or blue (decrease in copy number), density of genes with a positive dN/dS ratio (dN/dS) and locations of runs of homozygosity (ROH) highlighted in black. Each statistic is calculated across the 11 megabase chromosomes for each monomorphic clade. CNV (outer circles) and ROH (inner circles) are split into tracks for each clade ordered out to in; *T. b. evansi* type A (green), *T. b. evansi* type B (red), *T. b. evansi* type IVM-t1 (purple), *T. b. equiperdum* type OVI (yellow) and *T. b. equiperdum* type BoTat (orange).

stumpy formation, which is reduced or lost in monomorphic lineages exposed to BHI (Fig. S2)[4].

Of genes with a known involvement in QS signalling, replacement of pleomorphic alleles of Tb927.11.6600 (Hyp1), Tb927.8.1530 (TbGPR89) and Tb927.4.3650 (PP1) with monomorphic sequences did not alter developmental competence (Fig. S3b–d), although there was slow growth in parasites expressing the *T. b. equiperdum* type BoTat TbGPR89 variant (Fig. S3b). In contrast, a developmental phenotype was observed with some monomorphic clade-specific Tb927.2.4020 alleles (Figs. 2a and S3a). This gene encodes the NEDD8-activating enzyme E1 (APPBP1), which along with UBA3, binds NEDD8 and activates protein neddylation, a highly conserved eukaryotic posttranslational modification pathway required for diverse cellular processes[30–34]. Monomorphic clade-specific nonsynonymous mutations in APPBP1 are found in *T. b. evansi* type A (*n* = 4), *T. b. evansi* type IVM-t1 (*n* = 5) and *T. b. equiperdum* type BoTat (*n* = 4), although the structural consequences of each were predicted by AlphaFold (*26*) to be minimal (Fig S4a, b). In standard media (HMI-9), cells expressing APPBP1 sequences from the monomorphic clades *T. b. evansi* type A and *T. b. equiperdum* type BoTat, control pleomorphic (TREU927/4) or pleomorphic add-back displayed no significant difference in growth over 72 h, although there was growth delay in the cell line expressing the *T. b. evansi* type IVM-t1 monomorphic APPBP1 sequence at 48 h (Figs. 2a and S3a). In the presence of the QS oligopeptide signal, BHI, parasites expressing the *T. b. equiperdum* type BoTat and *T. b. evansi* type A sequences arrested growth equivalently to those expressing the pleomorphic *T. b. brucei* alleles (Fig. S3a). In contrast, cells expressing the *T. b. evansi* type IVM-t1 APPBP1 alleles continued to grow when exposed to BHI (Fig. 2a), indicative of reduced sensitivity to the QS signal. This correlated with a higher proportion of cells with 2 kinetoplasts and one or two nuclei (indicative of cells in active replication) and the absence of expression of the stumpy stage-specific protein, PAD1, in contrast to cells expressing the *T. b. brucei* sequence (Fig. 2b). Confirming the phenotype was dependent upon the monomorphic APPBP1 sequence, add-back of the pleomorphic *T. b. brucei* sequence to replace the *T. b. evansi* type IVM-t1 sequence in the cell line restored growth arrest and PAD1 expression once exposed to BHI (Fig. 2a, b).

Interestingly, despite their contrasting phenotypic effects, the *T. b. evansi* type A and *T. b. evansi* type IVM-t1 APPBP1 sequences differ by only a single nonsynonymous mutation, G224S (Fig. S4a). A cell line was generated that expressed APPBP1 with a sequence identical to pleomorphic *T. b. brucei* but with the single G224S mutation specific to *T. b. evansi* type IVM-t1. These cells arrested their growth upon exposure to BHI equivalently to those expressing the pleomorphic *T. b. brucei* sequence (Fig. S3e). Hence, the G224S mutation alone is not sufficient to reduce developmental competence but operates in the context of the other mutations in the *T. b. evansi* type IVM-t1 APPBP1 sequence.

Of those genes not previously linked to the QS response but identified based on their evidence for selection, Tb927.5.2580 has a positive dN/dS ratio in the monomorphic clade *T. b. evansi* type A. The sequence contains four clade-specific nonsynonymous mutations compared to the genome reference strain TREU927/4 (Fig. S4f) and encodes a mitochondrially-located 'MIX'-associated protein that cofractionates associated with the cytochrome c oxidase complex (complex IV)[35] which is predominantly expressed in procyclic forms[36,37]. Interestingly, cells expressing the *T. b. evansi* type A Tb927.5.2580 sequence grew more rapidly in HMI-9 media compared to cells expressing the *T. b. brucei* sequence (Fig. 2c) and, in the presence of the QS signal, continued proliferation and displayed fewer PAD1+ cells contrasting with cells expressing the pleomorphic sequence (Fig. 2c, d). Supporting its contribution to development, homozygous add-back of the *T. b. brucei* sequence caused parasites to reduce their growth rate in HMI-9 media and restored arrest after exposure to BHI (Fig. 2c). This was accompanied by reduced 2K1N,

2K2N cells and increased PAD1 expression although not fully to wild type levels likely due to the number of transfection cycles (Fig. 2d).

Confirming these in vitro assays, analysis in vivo demonstrated that parasites with the *T. b evansi* type A Tb927.5.2580 sequence showed reduced arrest and stumpy formation compared to parasites expressing the pleomorphic alleles and re-introduction of the wild type allele reversed this (Figs. 3 and S5). In the *T. b. evansi* type A clade, Tb927.5.2580 contains one nonsynonymous homozygous mutation which is unique, A149P. Expression of the *T. b. brucei* pleomorphic sequence with just the A149P mutation did not impact developmental competence (Fig. S3f). Hence, as for the G224S mutation in APPBP1, A149P appears to take effect in the context of other changes in the gene sequence rather than in isolation.

Analysis of monomorphic-specific alleles that also displayed a developmental phenotype in a high throughput *T. brucei* phenotypic screen[38] highlighted Tb927.11.3400 which encodes FAZ41, a component of the trypanosome flagellum attachment zone (FAZ). *T. b. equiperdum* type OVI and *T. b. equiperdum* type BoTat share five identical nonsynonymous mutations (Fig. S4j). Although each is believed to use sexual transmission, these isolates are of independent origin, phylogenetically separated by the pleomorphic TREU927/4 isolate (Fig. 1a). Replacement of FAZ41 with the *T. b. brucei*, *T. b. equiperdum* type BoTat and *T. b. equiperdum* type OVI sequence did not alter developmental competence (Fig. 2e). However, cells expressing either the *T. b. equiperdum* type OVI or *T. b. equiperdum* type BoTat sequence displayed significantly reduced motility in comparison to those expressing the *T. b. brucei* sequence. Add-back of the pleomorphic sequence restored motility to both initial replacement cell lines (Fig. 2f; Supplementary movies 1–5). Given that monomorphic *T. b. evansi* (unspecified type) show altered motility as a potential adaptation to the tissue environment[39], this highlights that our genetic comparisons can identify known phenotypic changes linked to monomorphism beyond developmental mechanisms. This was further corroborated by identification in our genome analyses of the distinct F1 ATPase γ subunit (Tb927.10.180) mutations that permit kDNA loss previously identified in some monomorphic clades[23]. A full list of genes with a clade specific high-impact or moderate-impact mutation in a monomorphic clade along with a dN/dS ratio <=1 in pleomorphic isolates and >1 in any monomorphic clade is provided in Supplementary Data 2, with predicted gene and protein alignments for all genes from all isolates accessible via GitHub (https://github.com/goldrieve/Mechanisms-of-life-cycle-simplification/blob/master/alignment_fasta/alignment.fasta.tar.gz). This provides a resource to explore other adaptations to non-tsetse transmission beyond the life-cycle simplification studied here.

## Gene regulators are downregulated as monomorphism develops

The presence of multiple co-dependent and context-specific mutations in different genes and clades, which impact developmental competence, could act to lock parasites in the monomorphic state but are unlikely to be responsible for the original progression toward monomorphism. To explore how monomorphism might arise in the first instance, we used a laboratory selection protocol to isolate developmentally incompetent parasites de novo. Specifically, we independently selected monomorphic *T. brucei* cell lines via parallel serial passage of clonal lines in increasing concentrations of BHI (Fig. 4a), and then generated clones from the selected lines after 30 passages. The reduced developmental competence of each selected line was then validated in vitro via exposure to BHI (Fig. 4a). In each case a reduced QS response was observed.

Thereafter we analysed by the transcriptome profile of each clonal line before and after selection. This demonstrated that the pleomorphic progenitors and monomorphic descendants are clearly discriminated and clustered with respect to their developmental competence, indicating a common transcriptome trajectory among

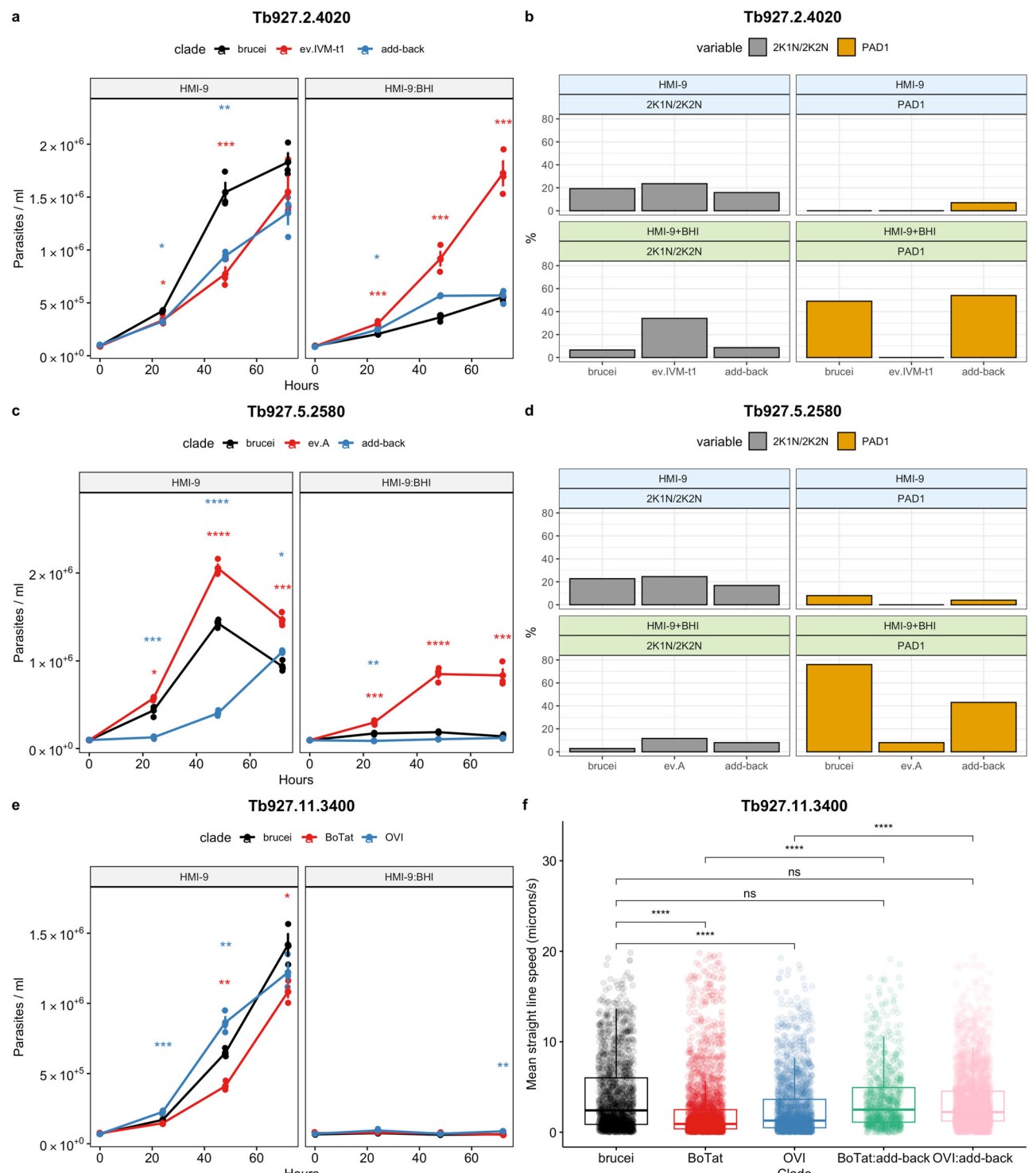

**Fig. 2 | Monomorphic *T. brucei* clades display mutations in discrete genes which hinder pleomorphic phenotypes. a, c, e** Growth and (**b, d**) percentage of the population replicating and PAD1 + % from pleomorphic *T. b. brucei* EATRO 1125 AnTat1.1 J1339 subjected to replacement of endogenous gene targets (**a, b**) Tb927.2.4020 APPBP1, (**c, d**) Tb927.5.2580−hypothetical protein and (**e, f**) Tb927.11.3400−Flagellum attachment zone protein 41 (FAZ41). The replacement cell lines were grown in HMI-9, or HMI-9 supplemented with an in vitro mimic of the QS signal, oligopeptide broth BHI (15%). Significance at each timepoint and for each comparison was tested using a repeated measures ANOVA, including an adjusted post-hoc Bonferroni test and significance indicated (*T. b. brucei* vs

monomorph, *; *T. b. brucei* vs add-back, *). * ($p < 0.05$); ** ($p < 0.01$); *** ($p < 0.001$); **** ($p < 0.0001$). For the growth and IFA analysis, four flasks were grown, three biological replicates for growth analysis and one to screen PAD1 and K/N counts at 48 h. **f** The motility of the FAZ41 replacement cell lines was also quantified for cells expressing the *T. b. brucei* ($n = 1758$), *T. b. equiperdum* type BoTat ($n = 2618$), *T. b. equiperdum* type OVI ($n = 2003$), *T. b. equiperdum* type BoTat:add-back ($n = 867$) and *T. b. equiperdum* type OVI:add-back ($n = 3192$) each compared between three biological replicates using a Wilcoxon two-sample test and the significance is indicated (* ($p < 0.05$); ** ($p < 0.01$); *** ($p < 0.001$); **** ($p < 0.0001$)). Source data are provided as a Source Data file.

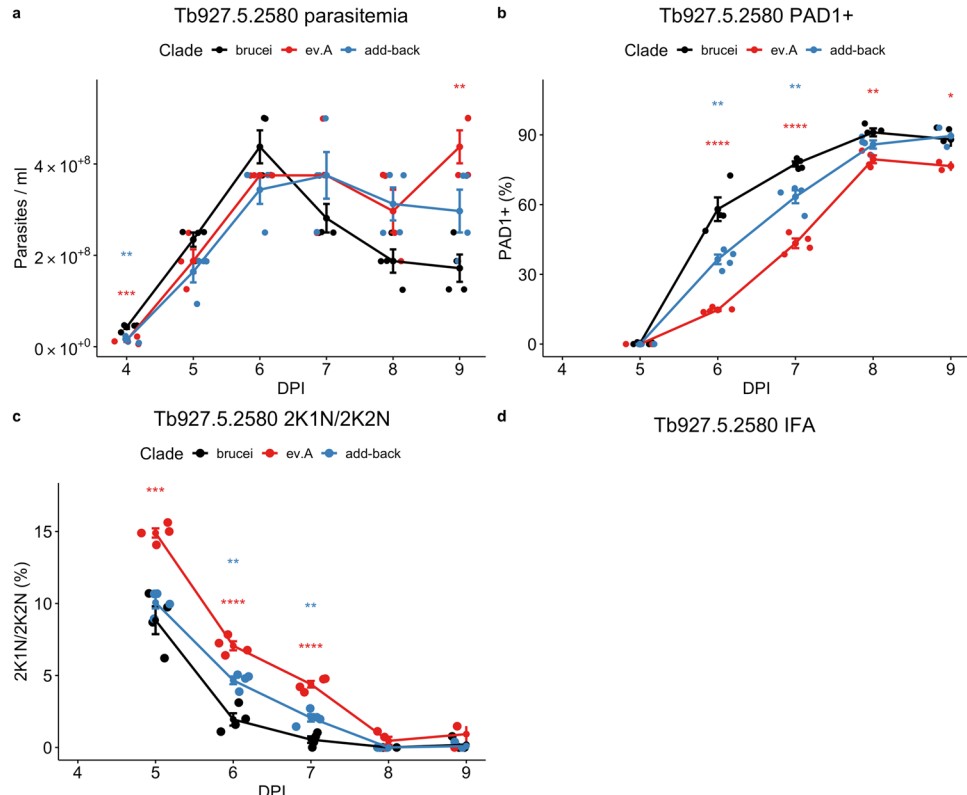

**Fig. 3 | Pleomorphic *T. b. brucei* expressing the monomorphic *T. b. evansi* type A Tb927.5.2580 sequence delays developmental progression. a** In vivo growth of pleomorphic *T. b. brucei* expressing the monomorphic *T. b. evansi* type A Tb927.5.2580 sequence or the pleomorphic *T. b. brucei* sequence, **(b)** cell cycle stage and **(c)** percentage of PAD1+ cells as assessed by immunofluorescence. Each cell line was used to infect four mice, represented by the dots at each time point.

Error bars represent mean standard error. Significance at each timepoint and for each comparison was tested using a repeated measures ANOVA, including an adjusted post-hoc Bonferroni test and significance indicated (brucei vs monomorph, *; brucei vs add-back, *). * ($p < 0.05$); ** ($p < 0.01$); *** ($p < 0.001$); **** ($p < 0.0001$). **d** Representative DAPI, PAD1 and merged images of each cell line at 6 DPI. Scale bar = 30 μm. Source data are provided as a Source Data file.

distinct selected lines; overall, the transition from pleomorphism to monomorphism accounted for 41% of the transcriptome variance (Fig. 4b). However, only 23 genes were significantly differentially expressed between all the selected clones (Fig. 4c, d). These genes were enriched for gene regulators, with GO terms including 'regulation of gene expression', 'posttranscriptional regulation of gene expression' as well as 'regulation of macromolecule metabolic processes' (Fig. S6d). Interestingly, the cohort of regulated transcripts included RBP10 (Tb927.8.2780), a known developmental regulator in trypanosomes[40] (Fig. S6a), which is itself predicted to bind[41] eight of the 23 commonly differentially expressed genes (Tb927.10.3410, Tb927.10.8050, Tb927.2.4200, Tb927.4.1910, Tb927.11.1280, Tb927.7.2680, Tb927.8.2780, Tb927.8.6490). Moreover, analysis of the expression of a manually curated list of quorum sensing (QS) genes (Fig. S6a) highlighted the key regulator ZC3H20 (Tb927.7.2660)[9,42], as well as PKA-R (Tb927.11.4610) and adenylosuccinate synthetase (Tb927.11.3650)[7] are differentially expressed during the selection for monomorphism (Fig. S6a).

We envisioned that the downregulation of RBP10 and its targets, alongside additional QS regulators such as ZC3H20, could initiate the first steps in the development of monomorphism, potentially reversibly in the first instance. To explore this, we re-expressed either RBP10 or ZC3H20 in a selected monomorphic line (A7) using a doxycycline-inducible expression system and assayed their responsiveness to the BHI QS signal. Fig. 4e demonstrates that inducible expression of either ZC3H20 or RBP10 in the selected monomorphic clone A7 resulted in growth and cell cycle arrest of each in response to BHI (Fig. S6e). The expression of these individual RNA regulators in the selected line was not, however, sufficient in isolation to restore PAD1 expression (Fig. S6f).

## Discussion

The generation of stumpy forms is maintained in *Trypanosoma brucei* populations to optimise transmission by tsetse flies[43]. However monomorphic isolates have been able to simplify their lifecycle by excluding the tsetse fly, allowing them to expand their geographic range via direct transmission between mammalian hosts. Here we have analysed the genomes of field isolates to identify molecular adaptations contributing to this simplified lifecycle, identifying mutations in known regulators and previously uncharacterized QS components, as well as changes in genes affecting growth and motility. Moreover, multiple mutations within individual genes affecting stumpy formation were observed, which in APPBP1 and Tb927.5.2580 were phenotypically co-dependent. A similar phenotype was observed for PAG3, which was found to be disrupted in some monomorphic clades[44]. As the deterioration of APPBP1, Tb927.5.2580 and PAG3 are clade specific, we suggest these mutations arose secondarily to the transition to monomorphism.

To explore the initiation of this phenotypic change, we performed laboratory selection to generate monomorphism de novo. In five independently selected clonal lines this generated consistent changes in the transcriptome, with key posttranscriptional regulators showing altered expression, including two with established roles in lifecycle progression (RBP10, ZCH20); previous selections for monomorphism have also identified changes in posttranscriptional regulators[45,46]. Interestingly, we show that in selected monomorphs ectopic re-expression of the regulators allowed restoration of sensitivity to BHI, demonstrating some reversibility of the phenotypic change to monomorphism in these newly selected lines. This resembles the sensitivity of *T. b. evansi* type A RoTat1.2 to BHI, where growth is inhibited without

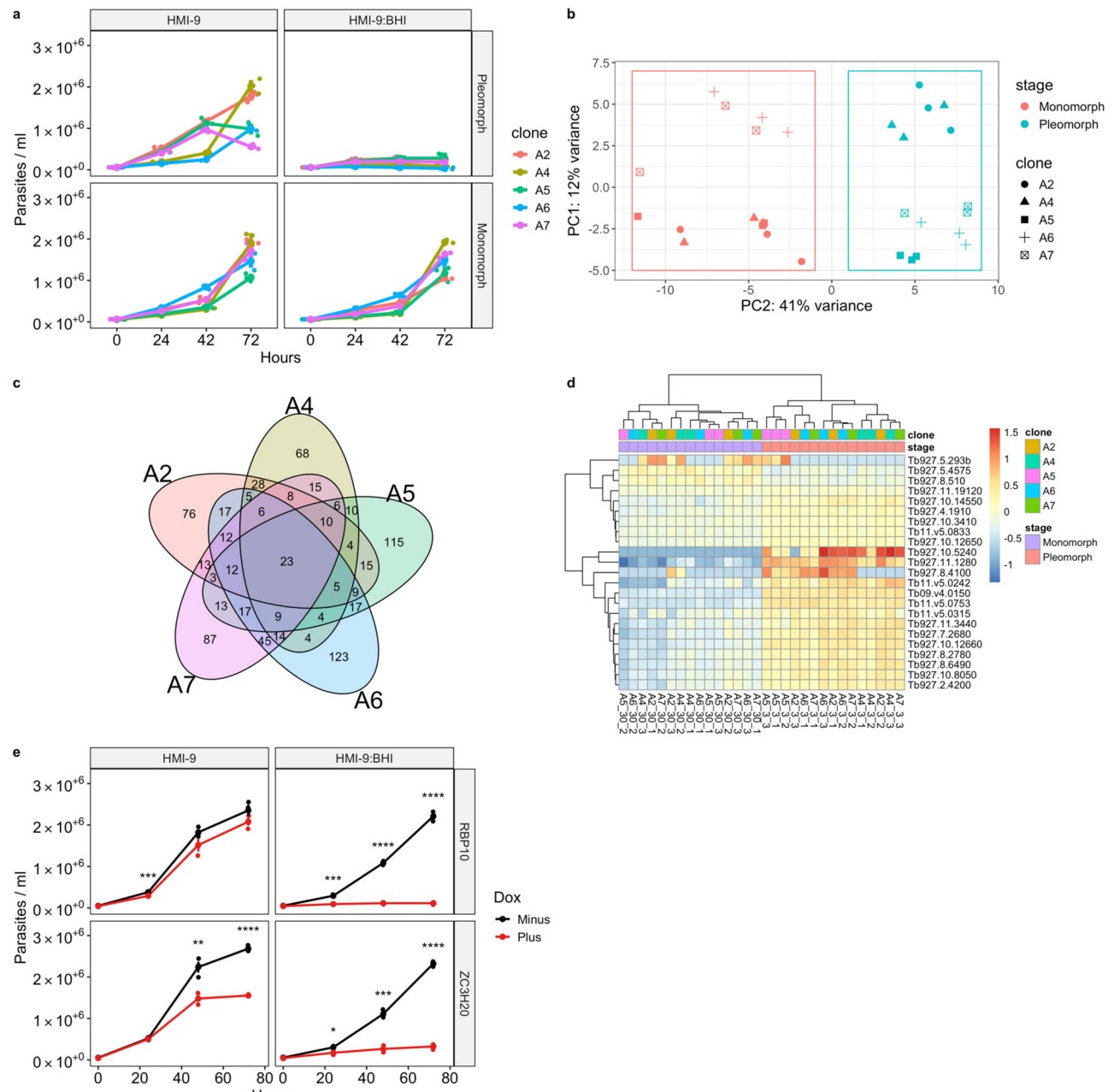

**Fig. 4 | Selected monomorphs display reduced developmental progression, which is reversible by inducible expression of key gene regulators. a** Five independent clones, derived by passage in increasing concentrations of BHI, display reduced developmental competence. The clones were grown in either HMI-9 or HMI-9 supplemented with 15% BHI; lines after passage 3 (pleomorph) or passage 30 are shown (monomorph). **b** PCA plot highlights the majority of variance in RNAseq libraries from the selection is explained by the selection from pleomorph to monomorph. **c** 23 differentially expressed genes are shared between each of the five clonal selections for monomorphism. **d** Heatmap of the 23 differentially expressed genes. The scale represents the row Z score, indicating the amount each gene deviates by in a specific sample compared to the genes mean expression across all samples. **e** Growth of monomorphic clone A7 induced with doxycycline (Plus) or not (Minus) to express transgenic RBP10 and ZC3H20. Each cell line was grown in triplicate biological replicates, represented by the dots at each time point. A triangle represents the mean for each time point. Error bars represent mean standard error. Significance at each timepoint and for each comparison was tested using a repeated measures ANOVA, including an adjusted post-hoc Bonferroni test and significance indicated (* ($p < 0.05$); ** ($p < 0.01$); *** ($p < 0.001$); **** ($p < 0.0001$). Source data are provided as a Source Data file.

PAD1 expression (Fig. S2), perhaps indicating an early stage of monomorphism in that clade.

In the field, we predict that monomorphism is initially reversible via changes in the expression of posttranscriptional regulators, with parasites ('proto-monomorphs') able to sustain transmission flexibility either through their tsetse vector or directly, potentially providing a selective advantage where tsetse transmission is challenging (Fig. 5). However, where tsetse become scarce through altered land use, tsetse control efforts, host migration or climate change[47], the proto-monomorphs continue to evolve as obligate asexual organisms through mutation. Muller's ratchet suggests that the organisms will be unable to purge these mutations through sexual recombination[17]. Eventually, the accumulation of mutations in key QS genes, independent gradual loss of their kDNA[18,48] and accumulation of mutations in the mitoproteome, ensures the loss of capacity for vectorial transmission and parasites become 'locked' as monomorphs. This has the short-

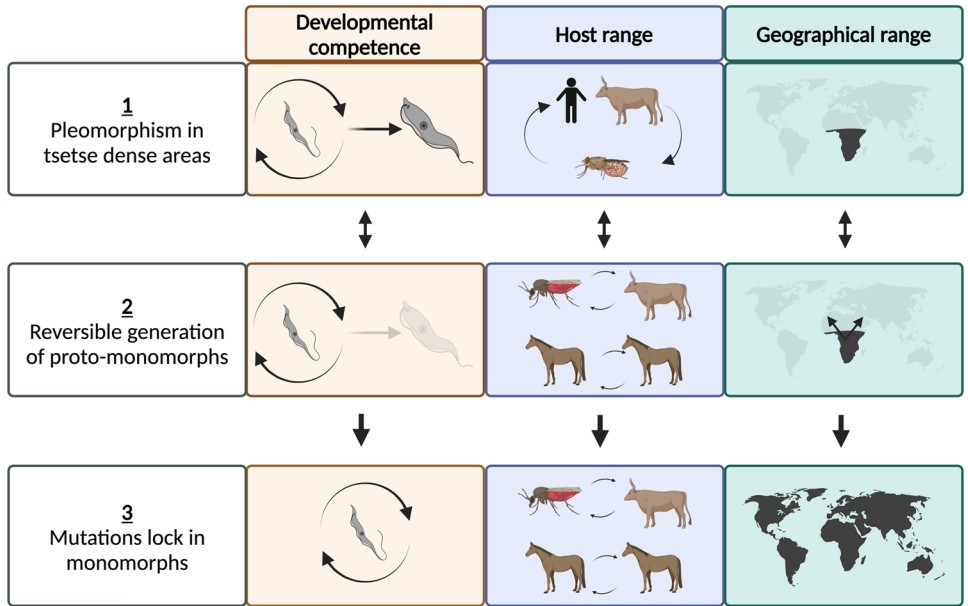

**Fig. 5 | A model for lifecycle simplification in *Trypanosoma brucei*.** This proposes that (1) in tsetse dense areas, developmental competence is maintained, supporting the genetic diversity of the parasite population through sexual recombination in the tsetse vector. (2) If tsetse numbers fall, or infected hosts move from a tsetse endemic area, transmission flexibility can be favoured by downregulating developmental regulators, forming proto-monomorphs, promoting enhanced parasitaemia and so mechanical transmission. (3) With prolonged mechanical transmission, mutations accrue which would eventually lock the proto-monomorphs into an obligate asexual/ monomorphic lifecycle. Created in BioRender (https://BioRender.com/t96e131).

term advantage of sustaining transmission in the absence of tsetse and allows the expansion of the geographic range of the parasites beyond the tsetse belt into Asia, South America, and Europe.

Whilst the accumulation of deleterious mutations would eventually doom the monomorphic clade to extinction, our data suggests that new monomorphic variants may arise frequently and without detection, given their initial flexibility in transmission mode, and prior to the emergence of secondary characteristics such as kDNA loss[13,18,23]. Given the drive toward enhanced virulence by adopting mechanical transmission[49], it will be important to detect emergent monomorphic clades where tsetse population control strategies and climate change are modifying the range of the vector[50], particularly for zoonotic *Trypanosoma brucei rhodesiense*[46,51]. The molecular events we have identified that underpin this process can also provide the necessary diagnostic tools[12] to detect and anticipate this threat.

## Methods
### Parasite cell lines
Publicly available genome data were accessed from NCBI (SRA)[52], the Wellcome Sanger Institute or generated as part of this study. Isolates sequenced during this study were propagated as bloodstream-form populations in OF1 mice (Charles River, Belgium) and purified from infected mouse blood via DEAE ion exchange chromatography; we recognize that some isolates described as pleomorphic in this analysis may have undergone adaptation which potentially hindered their developmental competence as part of their DNA isolation steps and prior to genomic analysis. Purified trypanosomes were sedimented by centrifugation (3,000 × g, 10 min at 4∘C) and DNA was extracted by standard phenol-chloroform. The concentration of extracted DNA was determined using a Qubit 4 Fluorometer (Invitrogen by Thermo Fisher Scientific). Paired end 150 bp sequences were generated using the DNA nanoball sequencing technology (DNBSEQ™) at the Beijing Genomics Institute (BGI). Animal experimentation procedures were approved by the Ethics Committee for Animal Experimentation (ECD-ITG) at the Institute of Tropical Medicine in Antwerp, under proposal number DPU-2017.

### Variant calling
The quality of the raw reads was analysed with FastQC[53] and subjected to quality trimming with Trimmomatic[54]. The trimmed reads were aligned to the *T. brucei* TREU927/4 V5 reference genome with bwa-mem[55]. The reads were prepared for variant calling following the GATK4 best practices pipeline, which included marking duplicate reads[56,57]. The variants were combined and filtered with stringent cut-offs, in keeping with GATK's best practices pipeline and previous studies[20,21]. The complete list of genomes analysed in this study, including their accession IDs, is summarised in Supplementary Data 1. Clade-specific mutations were assigned to annotated genes in the TRUE927/4 reference genome using snpEFF[58]. Protein domains and putative phosphorylation sites were identified in target genes using InterProScan accessed through TriTrypDB (www.TritrypDB.org). snpSIFT was then used to identify mutations which were specific to a monomorphic clade using the case-control function[58].

### Phylogenetic analysis
The filtered variants, described above, were filtered again to retain SNPs where a genotype had been called in every sample, using VCFtools[59]. A concatenated alignment of each variant was extracted using VCF-kit[60]. IQ-TREE[61] was used to create a maximum-likelihood tree from homozygous variant sites. Within the IQ-TREE analysis, a best-fit substitution model was chosen by ModelFinder using models that included ascertainment bias correction (MFP + ASC)[62]. ModelFinder identified TVM + F + ASC + R3 as the best fit and 1000 ultrafast bootstraps generated by UFBoot2[63]. The consensus trees were visualised and annotated with ggtree[64].

### Genome feature summary
vcf2fasta.py (available here https://github.com/santiagosnchez/vcf2fasta) was used to generate a fasta file of each gene for each isolate based on TREU927/4 gene annotations. Pseudogenes and VSGs were removed from the analysis. CodemI[65,66] was then used to calculate the dN/dS ratio for each monomorphic clade, using the variants and phylogenetic tree created above. Only genes with a perfect alignment

were included in the analysis, to avoid false positive results from misaligned sequences. The ctl file for each gene included: noisy = 3, verbose = 1, seqtype = 1, ndata = 1, icode = 0, cleandata = 0, model = 2, NSsites = 0, CodonFreq = 7, estFreq = 0, clock = 0, fix_omega = 0, omega = 0.5.

The dN/dS ratio was summarised for each clade and specific interest was taken for three groups of genes; (1) QS pathway genes, (2) genes used during the insect stage of the parasite's lifecycle, specifically those which were significantly differentially expressed between parasites extracted from the midgut/proventriculus stage and proventriculus/salivary gland stage[67], (3) essential genes, defined as those with a significant reduction in every library of the RITseq high throughput phenotype study[38]. The data was summarised in R.

Runs of homozygosity were identified with plink[68] using the following parameters: --homozyg-snp 20 --homozyg-kb 50 --homozyg-density 10 --homozyg-window-snp 10 --homozyg-window-missing 4 --homozyg-het 0 --maf 0.01 --homozyg-group --allow-extra-chr --family --write-cluster --cluster.

Dicey chop[69], bwa mem[55], samtools[70] and dicey map were used to create a mappability map for the TREU927/4 reference genome. CNVs were identified, merged and genotyped with delly[71] and outputs were merged again with bcftools[72]. Delly was used to classify each CNV and summarised with bcftools.

The density of 17,820 clade-specific variants (17,820 variants were chosen as this represents the lowest number of variants found in any of the clades), the density of genes with a positive dN/dS ratio, locations of ROH locations of CNV were visualised for each monomorphic clade with circlize[73].

Gene ontology (GO) enrichment (biological process) was calculated for genes in the *T. b. equiperdum* type OVI CNV and genes with positive clade specific dN/dS values using TriTrypDB. Only GO terms with a significant Benjamini value below 0.05 were reported (Fig. S1).

## Target prediction
Clade-specific variants described above were prioritised to create a target list of genes to validate their role in monomorphism. Initially, pseduogenes, VSGs and genes not on the megabase chromosomes 1-11 were removed. These genes were further filtered to create two target categories using the following criteria:

1. Genes which have a clade specific high-impact or moderate-impact mutation in a monomorphic clade in a known QS pathway gene[4,7,8,74]. The gene must also display no high impact mutations in any pleomorphic isolates.
2. Genes which have a clade specific high-impact or moderate-impact mutation in a monomorphic clade along with a dN/dS ratio <=1 in pleomorphic isolates and >1 in any monomorphic clade. The gene must also have a smaller log fold change in D3, D6 and PF than in the DIF category whilst displaying a log fold change in the DIF category greater than -1.5[38]. The gene must also display no high impact mutations in any pleomorphic isolates.

After filtering, 38 genes from Category 1 and 540 genes from Category 2. The full lists can be found in Supplementary Data 3. Six genes were taken forward for validation (four genes from Category 1 and two genes from Category 2).

## Protein structure prediction
Representative monomorphic mutant and pleomorphic control amino acid sequences were used to generate a protein structure model with ColabFold[75], which uses AlphaFold2[76] and Alphafold2-multimer[77]. The protein models were created using default settings and the best model, based on a predicted local-distance difference test (pLDDT), was aligned, and visualised with PyMOL[78].

## Synthesis of target replacement sequences
From our target list, we confirmed that all the target protein sequences were identical within the clade of interest and the sequence from a representative isolate from each clade was then chosen to synthesise a nucleotide sequence (Genewiz Gene Synthesis). As a control for each of these targets, the pleomorphic TREU927/4 sequence was also synthesised. To aid the downstream processing of the synthesised sequences, we added the enzyme sites HindIII and BamHI at the 5' and 3' terminus, respectively. HindIII and BamHI sites were identified within the coding sequence of Tb927.8.1530 and Tb927.2.4020, respectively. The sites were re-coded to remove the enzyme site whilst maintaining codon usage.

## Generation of plasmids required for CRISPR/ Cas9 transfection
The synthesised monomorphic sequences were generated in the pUC-GW-Kan plasmid (Genewiz) and transformed into XL1-competent cells. Plasmid preparation was performed using the GeneJET Plasmid Mini-prep Kit (Thermo Fisher), following the manufacturer's instructions. The plasmid was then digested using the HindIII and BamHI high-fidelity enzymes (NEB). The digested gene was excised from an agarose gel using the Monarch DNA Gel Extraction Kit (NEB). The gene was inserted into a pPOT plasmid using T4 DNA Ligase (NEB). Blasticidin (pPOT V6), hygromycin (pPOT V7), phleomycin (pPOT V7) and G418 (pPOT V7) resistance plasmids[79,80] using T4 DNA Ligase (NEB). Blasticidin, hygromycin, phleomycin and G418 resistance pPOT plasmids[80] were used at different stages throughout the following experiments, however, the same drug resistance plasmid was used for each set of experiments on a given gene. Hygromycin and Blasticidin were used to replace the wild-type alleles of every target gene apart from TbGPR89, for which we used Hygromycin and Phleomycin. Add-back of wild-type sequences to the initial replacement cell lines was performed using Phleomycin and G418. The following concentrations of each drug were used: G418 (2.5 μg/ml), Hygromycin (0.5 μg/ml), Puromycin (0.05 μg/ml), Blasticidin (2 μg/ml) and Phleomycin (2.5 μg/ml).

## Generation of products required for CRISPR/ Cas9 transfections
The Bar-seq primer design from LeishGEdit[80,81] was used to amplify a repair template from the pPOT plasmid. The LeishGEdit primers allow tagging of the wild-type allele or knock-out of the gene. Upstream and downstream, primers contain primer binding sites compatible with pPOT plasmids and 30 nucleotide homology arms for recombination. sgRNA, primers consist of a T7 RNA polymerase promotor (for in vivo transcription of RNA), a 20 nucleotide sgRNA target sequence to introduce the double-strand break at a locus of interest and a 20 nucleotide overlap to the sgRNA backbone sequence allowing the generation of sgRNA templates by PCR[79,80,82]. We modified the primers in the LeishGEdit protocol to allow us to replace the wild-type gene with a tagged monomorphic or pleomorphic sequence we had previously synthesised. We used the upstream forward primer 1 as per the LeishGEdit protocol and, for the downstream reverse primer, we used the start of primer 2 which binds the GS-linker of the pPOT plasmid and the tail of primer 7 which binds the 3' UTR of the target gene. All of the replacements were N terminally Ty tagged[83] and introduced under 3' UTR endogenous expression. The 5' sgRNA and 3' sgRNA primers were maintained as per the LeishGEdit protocol The LeishGEdit protocol uses the TREU927/4 genome to create primer binding sites. The binding sites of the primer sequences to the endogenous locus were visualised and the binding sites of each were screened for a mutation in the *T. b. brucei* EATRO 1125 genome, based on SNP calls. If necessary, primers were recoded to match the EATRO 1125 sequence. A full list of primers used in this study can be found in Supplementary Data 3.

The repair and sgRNA products were amplified for transfection using Phire Hot Start II DNA Polymerase (Thermo Fisher). The PCR was run for 30 cycles with the following conditions varying for the repair or

sgRNA amplification respectively: annealing temperature 65°C and 60°C, 3% DMSO and 0% DMSO and primer concentration 0.5 μM and 2 μM.

5 μg of each product was used for transfections into the *T. brucei* EATRO 1125 AnTat1.1 J1339 (J1339) pleomorphic cell line[4]. The J1339 cell line contains the pJ1339 plasmid derived from pJ1173 that carries a puromycin resistance marker, the Tet Repressor, T7 RNA polymerase and Cas9, which is expressed constitutively[80]. To transfect the parasites, we pelleted cells at 800 g for 8 min. The cell pellet was then resuspended in 100 μl of transfection buffer (90 mM sodium phosphate, 5 mM potassium chloride, 0.15 mM calcium chloride, 50 mM HEPES, pH 7.3)[84] and transferred to an Amaxa cuvette and transformed in an Amaxa Nucleofector® II using the CD4 + T cells Z-001 program. Following transfection, the cells were immediately transferred into 5 ml of pre-warmed HMI-9 and then transferred into 96 well plates and diluted 1:5, 1:25, 1:125, 1:625 using HMI-9. After 24 h of incubation, the cells were treated with drugs using the concentrations described above. Approximately 5–8 days post-transfection resistant clones became detectable and were transferred to fresh selective media[85].

gDNA was extracted from clonal transfectants using the DNeasy Blood & Tissue Kit with an RNAse A step (Qiagen). Successful replacements were then validated using the section of primers 1 and 7 which bind gDNA.

## In vitro developmental competence phenotype screen

Replacement clones were seeded at $1 \times 10^5$ cells/ml and grown in either HMI-9 or HMI-9 supplemented with 15% BHI in triplicate. The parasite population density was counted at 0, 24, 48 and 72 h using the Beckman Coulter Z2 Cell and Particle Counter.

A separate culture was used to generate immunofluorescence images of the gene replacement cell lines. The gene replacement clones were seeded at $1 \times 10^5$ cells/ml and left for 48 h. Cell cycle status and PAD1 expression were analysed by 4′,6-diamidino-2-phenylindole (DAPI) (100 ng/ml) and an anti-PAD1 antibody[86]. The cells were washed with vPBS and then resuspended in vPBS and 8% PFA (1:1) and incubated for 10 min at room temperature. 2 μL of IGEPAL CA-630 (Sigma-Aldrich) 10% in PBS was added and the cells were incubated for a further 10 min. The cells were centrifuged at 900 g for eight minutes and resuspended in 0.1% glycine in vPBS and incubated for 10 min. The cells were then centrifuged at 900 g for eight minutes and resuspended in vPBS. Fixed cells were added to a Polysine slide and left to dry. The cells were rehydrated in PBS for five minutes and blocked by incubation of the cells in PBS 2% BSA for one hour. The blocking buffer was removed and anti-PAD1 (1:1000) PBS 0.2% BSA was added to the cells and incubated for one hour. The cells were washed 5x with PBS and then incubated in the Alexa Fluor 568 Goat anti-Rabbit secondary antibody (1:500) (Invitrogen) PBS 0.2% BSA for one hour. 25 μL of DAPI (100 ng/ml) in PBS was added to the cells and incubated for five minutes. The cells were then washed five times with PBS. Finally, the slides were mounted with Fluoromount-G Mounting Medium (Invitrogen). All steps were performed at room temperature. The slides were then imaged with a ZEISS Axio Imager 2 using a 40x objective with a Prime BSI Camera. Images were generated with Micro-Manager 2.0[87] and analysed with FIJI[88]. The number of arrested (1K1N) and dividing (2K1N and 2K2N)[89] parasites were counted along with the number of PAD1+ cells.

The motility of FAZ41 replacement cell lines were quantified using $5 \times 10^5$ cells which were centrifuged at 800 g for five minutes and resuspended in 40 μl of HMI-9. The resuspended cells were then placed onto a slide and a cover slip was mounted using Vaseline. The slides were then incubated for 5 min (37°C, 5% CO2). Each slide was visualised at 40x magnification using the Olympus CKX53 and a time-lapse image was generated with a QImaging Retiga-2000R digital camera. 100 images were then taken, one image every 0.25 s. The images were imported into FIJI[88] and the background was subtracted (rolling ball

radius: 10, light background) from the images. The image was then inverted to improve the contrast between the cell and the background. TrackMate[90] was used to identify cells using default parameters, unless stated otherwise. LoG detector was used with estimated object diameter: 15 and quality threshold: 0. An initial spot threshold of 0.74 was then applied and the advanced Kalman Tracker to track cell motility (initial search radius: 30, search radius: 30 and max frame gap 20). A summary of each track was then exported for each time lapse.

For analysis of monomorphic cell line responsiveness to BHI, *T. b. brucei* EATRO AnTat1.1, *T. brucei* Lister 497, *T. b. equiperdum* type OVI (OVI isolate) and *T. b. evansi* type A (RoTat 1.2 isolate) were incubated with HMI-9 containing 0%, 1%, 2.5% or 5% BHI for 48 h. Cell counts were performed using a Neubauer slide (140468, Dominique Dutscher) after 48 h incubation with a Motic AE30 inverted binocular microscope (Motic, Hong Kong, China). Parasite solutions from HMI-9 medium (negative control) and HMI-9 medium supplemented with 5% BHI were used for PAD1 expression analysis. $2 \times 10^6$ trypanosomes were centrifuged at 900 g for 5 min. The culture supernatant was removed and the parasite pellet was washed with cold 1X PBS. The washed cells were then fixed with 4% paraformaldehyde (PFA) for 10 min on ice. Cells were resuspended in 130 μl of 1X PBS - 0.1 M glycine and incubated overnight at 4 °C. Cells fixed in 4% PFA were then diluted in 1X PBS to a concentration of $1 \times 10^4$ cells/ml. Twenty microliters of 4% PFA-fixed cells were deposited on an 8-well Labtek slide (Dominique Dutscher - 2515364) previously treated with Poly-L-Lysine (P4707 - Sigma Aldrich), and the fixed cells were incubated at room temperature for 1 h. Membrane permeabilization of parasites adhering to the Labtek slide was performed using a PBS - 1X Triton 0.1% solution for 2 min, followed by washing with 1X PBS. Saturation of nonspecific sites in Labtek slide wells was performed with PBS 1X - BSA 2% solution for 45 min at 37 °C in a humidity chamber prior to addition of 20 μl anti-PAD1 primary antibody (diluted in PBS 1X - BSA 2%). Wells were washed 5 times with PBS 1X and then incubated with 20 μl of secondary antibody (diluted in PBS 1X - BSA 2%, α-lapin Alexa fluor 488 1:500) for 45 min at 37 °C in a humidity chamber before incubation for 10 min with 50 μl of DAPI solution (NucBlue Hoechst Live Cell Stain ReadyProbes, ThermoFischer - R37605). Wells were washed 5 times, then a drop of ProLong® Gold Antifade Reagents (Invitrogen™ P36941) was applied before mounting the slides. Slides were then analyzed on an Axiolab® Zeiss microscope using a 40x objective. Negative controls (no primary antibody and no secondary antibody) and positive controls (*T. brucei* EATRO AnTat1.1 incubated 48 h in HMI-9 medium with 5% BHI) were included for each experiment.

## Monomorph selection and transcriptomic profiling

Pleomorphic *T. brucei* EATRO 1125 AnTat 1.1 90:13 was cloned via serial dilution. In a similar approach to previous studies, the cell line was grown in HMI-9 at 37 °C and 5% $CO_2$[91] for a total of 72 days[45]. In this study, we supplemented the HMI-9 with BHI. During the 72-day selection, the percentage of BHI was progressively increased from 2.5% to 15%. The cell density was counted at each passage using a Beckman Z2 Coulter particle counter and size analyser. High cell density was deliberately maintained to select proliferating cells at high parasite population density. Cultures were cryopreserved weekly during the selection period at −80 °C in HMI-9 supplemented with 10% glycerol. Before the cultures were screened for their responsiveness to BHI, the cells were washed twice with PBS and grown in HMI-9 for one week to remove any residual BHI from the media.

RNA was extracted from triplicate cultures of the 'start' and 'end' from the five clones described above. The cells were grown to a density of between $7.5 \times 10^5$ cells/ml and $1 \times 10^6$ cells/ml and RNA was then extracted using the Qiagen RNeasy Mini kit, following the manufacturer's instructions. RNA was then sequenced with DNBseq (~4 Gb/sample) by BGI (Hong Kong).

Our analysis was based on a standard workflow for analysing RNAseq data[92]. Salmon[93] was used to index the TREU927/4 reference genome, which was then used to quantify transcript abundance for each sample. Transcript abundance values were imported into R with tximport[94]. A database was created using GenomicFeatures[95]. Genes with a count of less than 10 were excluded from the analysis and a VSD transformation was applied to visualise the data as heatmaps and PCA plots. Clones 1 and 3 were removed from the analysis as the replicates did not group via PCA.

DESeq2[92] was used to quantify differentially expressed genes. Differentially expressed genes were identified between the start and end of the selection. For the clonal selection, initially differentially expressed genes were identified by grouping all the clonal libraries. The contrast function was then used to identify differentially expressed genes for each clonal selection using the same model. To compare the similarity in differentially expressed genes between the clonal selections, a Venn diagram was created with Venn (available here https://github.com/dusadrian/venn) to highlight shared differentially expressed genes between the clonal selections. GO enrichment was calculated with TriTrypDB[96], including a Benjamini false discovery rate cut-off of >0.05. GO enrichment was visualised with ggplot2 in R.

### Overexpression of ZC3H20 and RBP10
ZC3H20 (Tb927.7.2660) and RBP10 (Tb927.8.2780) were chosen to attempt rescue pleomorphism in the selected monomorphic *T. brucei* cell line. The genes were amplified from pleomorphic *T. brucei* EATRO 1125 genomic DNA. These amplicons were ligated into the TOPO plasmid using T4 ligase (NEB), following the manufacturer's instructions. The ligated plasmids were then transformed into competent cells and grown overnight. A small-scale plasmid preparation was performed on the culture to isolate the plasmid using the GeneJET Plasmid Miniprep Kit (Thermo Fisher). This plasmid was digested using XbaI and BstXI (5' tag) or HindIII and SpeI (3' tag) enzymes (NEB) following the manufacturer's instructions, and the genes were ligated into the doxycycline-inducible pDex-577y plasmid[97]. This plasmid integrates into the 177 bp repeat sequences of minichromosomes and encodes for phleomycin resistance. ZC3H20-pDEX-577y and RBP10-pDEX-577y were transfected into selected monomorphic clone A7.

The developmental competence of the cell lines transfected with ZC3H20-pDEX-577y and RBP10-pDEX-577y was determined in vitro using BHI-based oligopeptides as above whilst gene overexpression was either uninduced or induced. Doxycycline induction was titrated from 0.2 ng/ml to 0.02fg/ml in 1:10 increments.

### In vivo infections
Experiments were carried out after receiving ethical approval according to UK Home Office Animal (Scientific Procedures) Act (1986) under licence number PP2251183. All experiments were performed using female, aged matched (>10 weeks), MF1 mice (*Mus musculus*). All animals were reared within the animal facility of the Ashworth Laboratories, University of Edinburgh. Mice were housed in Techniplast Blueline 1284 IVC cages within the animal facilities of the Ashworth Laboratories, University of Edinburgh (Home Office Establishment number X212DDDBD). A cycle of 12 h dark/ 12 h light was maintained with an ambient temperature of 21 °C and 55% humidity. In vivo infections were performed using four female MF1 mice per cell line. Mice were cyclophosphamide treated (25 mg/ml) 24 h before infection with 10,000 parasites via intraperitoneal injection. Parasitaemia was monitored from 3 days post-infection by tail snip. Blood-smeared slides were air dried at room temperature for five minutes and then fixed via submersion in methanol at −20°C for 10 min or stored in methanol at −20°C. PAD1+ and dividing cells were identified using the immunofluorescence protocol described above.

### Statistical analysis
A Wilcoxon two-sample test (Fig. 2f) or a repeated measures ANOVA, including an adjusted post-hoc Bonferroni test (Figs. 2a, c, e, 3a, b, c and 4e) was performed using Rstatix[98]. The data was summarised and checked for normality and outliers before analysis. Full details of statistical results can be found in the Source Data file.

### Reporting summary
Further information on research design is available in the Nature Portfolio Reporting Summary linked to this article.

## Data availability
All data are available in the main text, the Supplementary Materials, or NCBI under the BioProject PRJNA1114649. Source data are provided with this paper.

## Code availability
All code written for this study has been deposited on GitHub (https://github.com/goldrieve/Mechanisms-of-life-cycle-simplification) and published on Zenodo https://doi.org/10.5281/zenodo.14012494[99].

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

## Acknowledgements

We thank Dr. Frederik Van den Broeck for facilitating data sharing, Prof. Achim Schnaufer, Prof. Philippe Büscher, Dr. Joanna Young, Prof. Steve Kelly, Dr. Stephen Larcombe and Dr. Al Ivens for their valuable insights and comments. The work was funded by the following awards: Wellcome Trust (221717/Z/20/Z) K.R.M., Wellcome Trust (220058/Z/19/Z) - G.R.O., K.R.M., Wellcome Trust (108905/Z/15/Z) - F.V. and K.R.M., Medical Research Council Career Development Award [MR/W026996/1]. This UK funded award is carried out in the frame of the Global Health EDCTP3 Joint Undertaking - M.C., Bill & Melinda Gates Foundation (grant

number OPP1174221) and the Flemish Government EWI SOFI-2018 "Cryptic human and animal reservoirs compromise the sustained elimination of gambiense-human African trypanosomiasis in the Democratic Republic of the Congo" - M.G. and N.V.R. and GIS CENTAURE Recherche Equine and the Regional Council of Normandy - M.V. and L.H.

## Author contributions

Conceptualization: G.R.O., K.R.M. Methodology: G.R.O., F.V., M.C., M.V., L.H., M.G., N.V.R., K.R.M. Investigation: G.R.O., F.V., M.V., L.H., K.R.M. Visualization: G.R.O., K.R.M. Funding acquisition: L.H., M.G., N.V.R., K.R.M. Project administration: G.R.O., F.V., K.R.M. Supervision: M.C., K.R.M. Writing—original draft: G.R.O., K.R.M. Writing—review and editing: G.R.O., F.V., M.C., M.V., L.H., M.G., N.V.R., K.R.M.

## Competing interests

The authors declare no competing interests.
