## [Peer Review File · Nature Communications]

REVIEWER COMMENTS

Reviewer #1 (Remarks to the Author):

This is an interesting further addition to the differentiation story in trypanosomes, and has some novelty in that strian sequence data are exploited to indentify possible factors involved in progression between mammalian and insect infetious forms.

My cmmnts on specific things are below, but I think I have three major questions: How does this advance understanding, as as presented the data do not seem to mesh very well into a mechanistic model? How robust are the data? This is difficuat as the stats are ommitted from teh genetics analysis. And finally, what alternate hypothesise could explain these observations? We need to have some sort of Occum's razor here, as what is offered is adherence to a specific view.

Abstract is very coy - no details given. Would like to know what these changes are, at least in general. I had a similar feeling with the rest of the MS, which made this a tough read.

My understanding was that an enabling set of mutations have been described for kDNA loss? Is this not corerect? This is then mentioned in the deiscussion

Classical concept of a species has been questioned quite recently and perhaps is worth mentioning?

I fundamentally disagree that an asexual lifecycle prevents purging deleterious mutations. This is what selection is for! It also does not prevent adaptation or the aquisition of diversity. Much of this will be neutral, followed by selection. This is also contradicted in the discussion.

Odd prganisation as we are suddenly given data with no information on how obtained, in what I assume is the introduction?

Depending on the depth of sequencing SNPs can be hugely overestimated, especially with short reads. How has this been controlled? I'm a bit unclear as well as to why the phylogeny is based on SNPs and not full genome data? Is this simply a diversity phylogeny (not sure what the correct term would be here) rather than a true one? BUT, do we not already know that these strains have distinct clade structure? nHow des this new analyis compare?

It may be my ignorance but I am not clear that you can equate nonsynonymous mutations exclusively with deleterious mutations. I'm also unclear if the binning of data with vastly different numbers of strains (which I assume is the case) can be a major impact here, i.e. one clade has pneumonia strain, another has over ten. There is a major hotspot of ChrII and others - what do these represent?

Is it not expected that mitochondrial components will be sigificantly affected, but if these are neutral and have occurred independantly that these would differ?

There is no real statistical analysis, for SNPs, genome feaitres or the prevalence of GO terms.

ROH are cited as a possible feature to purge deleterious mutations, in contradiction to the statement in the introduction.

with cell cycle development and expression of the stumpy-specific marker protein PAD1

PAD1 can be upregulated by stress, and I am unclear as to what 'cell cycle development' means.

Need to be clearer as to what the changes made to the selected genes are. Usual to call these out as, eg, G259Y or similar rather than 'clade' which I cannot see as accurate. Also, helpful if the accessions were accompanied by the annotation for that gene. Do the mutations lead to destabilisation, truncation or? All three seem to have been implicated in differentiation by an earlier high throughput screen. This should be mentioned; it adds some weight to the arguments here.

Given that the three genes analysed are involved in vastly distinct processes, does seem a need for a unifying concept here., Also, as 'brucei' is odd that there are such differing responses. The lethal impact of BHI on all of the 927.11.3400 is spectacular, but how reconcile with life cycle progression in *T. brucei* itself?

Why only Tb927.5.2580 analysed in more detail?

Reviewer #2 (Remarks to the Author):

The group of trypanosomes *T. brucei sensu lato* has a number of peculiar features and one of them is the ability of some strains to eliminate the infamous tse-tse fly vector from their life cycle, with major consequences for their spreading outside the so-called tse-tse belt. While this phenomenon was recently studied from several perspectives, the authors of this paper took a new one, which this reviewer appreciates.

They focused on mutations potentially associated with the loss of stumpy formation, changes in quorum sensing and PAD1 expression, as factors that may play critical role in the switch from polymorphism to monomorphism (the involvement of flagellar mobility in this context is another novelty, at least for this reviewer). Using a set of *T. brucei equiperdum* and *T.b. evansi* strains (probably the most comprehensive available) as well as experimental reversion of *T. b. brucei* to monomorphism, and - importantly - expression of the potentially most interesting genes from the dys/akinetoplastic trypanosomes in *T. b. brucei*, and the rescue thereof. Combination of these approaches allowed them to identify mutations in known and novel components of quorum sensing, and other genes worth follow-up studies. The results are not black & white, as the individual identified mutations have either different importance in different strains/ecotypes, or work in tandem with other identified mutations or some yet unknown ones (some of the add-backs worked "half way", but that's not unusual for add-backs in *T. brucei*). Moreover, there are apparently various paths to monomorphism, which overlap only to some extent. But overall the

progress achieved by this study qualifies it, in the eyes of this reviewer, for Nat. Commun.

I am pretty happy with the experimental setup which is elegant, and presentation of the data, but I have a number of suggestions to be considered (especially for the introduction and discussion), as well as some omissions to be fixed.

58 - They start with the subspecies status, but it was shown recently that in the case of *T. brucei*, these are not subspecies. Even if they have hard time to accept that, they should at least mention that another alternative (a way better one in the view of this reviewer) has been published in Trends in Parasitology recently, namely that of the ecotypes.

71 - be concise - lifecycle versus life cycle (73), geographic vs geographical, nonsynonymous vs non-synonymous etc.

74 - a reference to the Lun paper (Trends in Parasitology) would be appropriate here

78 - biting flies sounds a bit inaccurate, tabanid flies would be better but if they are aware of any non-tabanids involved, then it is fine as is

93 - again - types, subspecies, ecotypes...

105 - reference to the review paper in TiP 2022, where this was shown very convincingly would be in place here

140 - I missed in the discussion any comments/elaboration on the accumulation of mutations in the mitoproteome

151-153 - any idea why these GO terms were overrepresented?

216 - perhaps appropriate mentioning here that complex IV subunits are present, albeit at low level, in bloodstreams (Zikova et al., PLoS Pathog. 2017).

235 - a high throughput - typo

335 - Muller's ratchet is correct

336-339 - it puzzles me why for the support of their view of gradual switch to monomorphism they do not cite paper(s) that show exactly that on the level of kDNA?!

342-344 - they may strengthen their interpretation of their data by including reference to a paper from Zhao-Rong Lun and collaborators, who already predicted the continuous emergence of new monomorphic strains out of Africa.

some small issues with references - double check Hoare 1970, missing number of the paper for 24,

errors in 27, 29 and probably others

REVIEWER COMMENTS

Reviewer #1 (Remarks to the Author):

This is an interesting further addition to the differentiation story in trypanosomes, and has some novelty in that strian sequence data are exploited to identify possible factors involved in progression between mammalian and insect infetious forms.

Thank you for these comments. We have answered all comments to the best of our ability below, although we were uncertain of the meaning of some comments where the context was not always defined or the meaning not fully clear in the written text of their review. Nonetheless, we hope we have addressed the queries raised and explained the methodological approaches we used, which are well established in the field. We hope this provides a better explanation of the approach to the reviewer and other readers who are perhaps less familiar with such analyses.

My cmmnts on specific things are below, but I think I have three major questions:

How does this advance understanding, as as presented the data do not seem to mesh very well into a mechanistic model?

We present the most comprehensive analysis on the independent evolution of monomorphic trypanosome clades to date. We show that each clade has features at the genome scale indicative of independent obligate asexual evolution. We then identify clade specific mutations which impact pleomorphic phenotypes, such as developmental competence and motility, and validate the impact of specific exemplars of these mutations experimentally.

It is, of course, impossible to identify the exact mutation that originally gave rise to the monomorphic phenotype in each clade as these mutations could (likely did) accrue after the organism had transitioned to an obligate asexual form, and therefore, would be subjected to the progressive accumulation of mutations.

Therefore, we also investigated the initial changes that led to the loss of developmental competence by selecting monomorphic isolates in vitro de novo and validated the role of key gene regulators in the initial development of monomorphism. This allowed us to generate an overall mechanistic trajectory for the evolution of monomorphism, detailed in the manuscript discussion. Moreover, we dedicated an entire figure to this model (Fig. 5) and our model was also nicely summarised by reviewer 2 indicating it was clearly articulated.

Specifically, we stated:

“In the field, we predict that monomorphism is initially reversible via changes in the expression of posttranscriptional regulators, with parasites ('proto-monomorphs') able to sustain transmission flexibility either through their tsetse vector or directly, potentially providing a selective advantage where tsetse transmission is challenging (Fig. 5). However, where tsetse become scarce through altered land use, tsetse control efforts, host migration or climate change [48], the proto-monomorphs continue to evolve as obligate asexual organisms. Muller's ratchet suggests that the organisms will be unable to purge these mutations through sexual recombination [17]. Eventually, the accumulation of mutations in key QS genes, and the loss of their kDNA [49], ensures the loss of capacity for vectorial transmission and parasites become 'locked' as monomorphs.”

How robust are the data? This is difficuat as the stats are ommitted from teh genetics analysis.

This comment is addressed in detail later in our response to the Referee's specific queries.

And finally, what alternate hypothesise could explain these observations? We need to have some sort of Occum's razor here, as what is offered is adherence to a specific view.

It is tricky to answer this concern without any context. Does reviewer 1 mean our interpretation of the results as a whole? The genome comparison? The laboratory validation of these mutations? The analysis of selected monomorphism?

Our manuscript presents our interpretation of the data, which is based on a combination of our robust experimental outcomes and the established background literature concerning the evolution of monomorphic *T. brucei*. We will run through potential null hypotheses/conclusions for each figure to address what we believe to be the Referee's queries

Fig. 1.

- Our conclusions: Monomorphic *T. brucei* are polyphyletic and form at least four independent clades, each displaying clade specific features at the genome scale
- Alternative conclusions: The sequencing data, and analysis, is not well supported by the data and therefore the clades are not polyphyletic and do not display unique features.

This alternative interpretation would contradict the existing literature on monomorphic phylogeny using genome data, where there is a good support for a polyphyletic origin of monomorphic clades. Our analysis using all available isolates is consistent with the polyphyletic origins of different lineages and was necessary to allow us to interrogate individual lineages for genomic features that may contribute to life cycle simplification, which were subsequently experimentally analysed.

Fig. 2:

- Our conclusions: Monomorphic *T. brucei* clades display mutations in discrete genes which alter phenotypes to favour the monomorphic lifestyle.
- Alternative conclusions: The identified mutations do not impact the pleomorphic phenotype

Although many mutations were identified and many do not demonstrably impact on developmental competence, our prioritisation strategy allowed us to identify genes with mutations that alter the pleomorphic phenotype, this being confirmed by experimental validation of their impact

Fig. 3:

- Our conclusion: Pleomorphic *T. b. brucei* expressing the monomorphic *T. b. evansi* type A Tb927.5.2580 sequence delays developmental progression in vivo
- Alternative conclusion: None

We demonstrate that parasites show reduced stumpy formation when the mutant alleles are expressed in pleomorphic parasites, confirming the contribution of these gene mutations to loss of stumpy formation in vivo as well as in BHI medium in vitro (shown in Figure 2),

Fig. 4:

- Our conclusions: Selected monomorphs display reduced developmental progression, which is reversible by inducible expression of key gene regulators.
- Alternative conclusions: None
 - We demonstrate that re-expression of the RNA regulators that show reduced expression in selected monomorphic parasites restores cell cycle arrest upon exposure to the BHI quorum sensing signal.

Fig. 5:

- Our conclusions: A model for lifecycle simplification in *Trypanosoma brucei*. This proposes that (1) in tsetse dense areas, developmental competence is maintained, supporting the genetic diversity of the parasite population through sexual recombination in the tsetse vector. (2) If tsetse numbers fall, or infected hosts move from a tsetse endemic area, transmission flexibility can be favoured by downregulating developmental regulators, forming proto-monomorphs, promoting enhanced parasitaemia and so mechanical transmission. (3) With prolonged mechanical transmission, mutations accrue which would eventually lock the proto-monomorphs into an obligate asexual/ monomorphic lifecycle

This model provides a framework to explain the origin of monomorphic parasites, which is initially reversible and associated with the downregulation of RNA regulators, after which parasite lock-in to the monomorphic phenotype through accrued mutations. This model is consistent with our data and previous models that have discussed the origin of monomorphic lineages. As with all models, other interpretations may exist but we believe that this provides a suitable explanation with existing information. Of course, as more knowledge becomes available through the analysis of even more monomorphic isolates or the experimental analysis of more identified mutations, the model may become further consolidated, refined or require modification.

Abstract is very coy - no details given. Would like to know what these changes are, at least in general. I had a similar feeling with the rest of the MS, which made this a tough read.

Thank you- although the abstract length is restricted to around 150 words, we have introduced some more detail. We have also expanded some elements of the manuscript text and format – we apologise that the submitted manuscript was rather concise which was necessary to conform to the article length and format policies of the journal on first submission.

My understanding was that an enabling set of mutations have been described for kDNA loss? Is this not correct? This is then mentioned in the discussion

It is correct that mutations that allow bloodstream forms to tolerate kDNA loss have been identified. However, kDNA loss occurs secondarily to monomorphism and is not a cause of the loss of stumpy formation in non-tsetse transmitted parasites (Dewar, C. E. et al. Mitochondrial DNA is critical for longevity and metabolism of transmission stage *Trypanosoma brucei*. PLoS Pathog 14, e1007195 (2018). <https://doi.org/10.1371/journal.ppat.1007195>).

Our manuscript sought to identify molecular changes that accompany or cause reduced development in the bloodstream. Many such changes were identified and for two exemplar genes a role in stumpy formation was confirmed experimentally by precise allelic replacement and add-back. However, all the described mutations that allow kDNA loss in distinct monomorphic lineages were also identified in our genomic analysis which provided excellent reassurance of our bioinformatic analysis. Our manuscript focused on changes that contribute to monomorphism but the accompanying data with the paper includes all changes identified in all monomorphic lineages, with DNA and protein alignments available for others to explore their own areas of interest. This data is available at the GitHub site associated with the manuscript.

Classical concept of a species has been questioned quite recently and perhaps is worth mentioning?

Thank you for raising this point (also raised by reviewer 2). Indeed, the nomenclature surrounding pleomorphic and monomorphic trypanosomes has been controversial, with several solutions having been proposed to resolve the complexity of differing phenotypes, genetic exchangeability, disease presentation and host specificity, transmission modes and the polyphyletic origins of monomorphic lineages. Among the most recent is the proposal of ecotypes. This is not yet universally accepted in the field as the favoured approach to resolve the issue, and there are complications when laboratory selection is used to alter characteristics that define ecotypes – as we have done when selecting monomorphic lines from pleomorphic lines in BHI where, for example, changes may be reversible/epigenetically controlled. Instead, we consider it is valuable to acknowledge the genetic relationships and phylogeny of different parasite lineages and so refer to subspecies and 'types' in our manuscript (which is also not without issues). Nonetheless, the introduction of ecotypes is a valuable proposal, and we are happy to acknowledge and cite the relevant publication, which also usefully debates the challenges with nomenclature for these parasites. Ultimately, the field as a whole will be required to find a consensus.

I fundamentally disagree that an asexual lifecycle prevents purging deleterious mutations. This is what selection is for! It also does not prevent adaptation or the acquisition of diversity. Much of this will be neutral, followed by selection. This is also contradicted in the discussion.

Thank you for raising this although we challenge this point and disagree that we contradict ourselves in the discussion. Nonetheless, we have reworded the manuscript text to clarify our meaning. The accumulation of deleterious mutations in obligate asexual organisms is an extremely well-studied and accepted phenomenon called Muller's ratchet, which is nicely summarised by Gabriel (1993)

"Muller (1964) pointed out the special severity of this problem for small asexual populations. Assuming back mutations are rare, in the absence of recombination, no individual can ever produce an offspring with fewer deleterious mutations than it carries itself. The possibility always exists that, by chance, the class of individuals with lowest fitness will not produce offspring in some generation. After this class of individuals has been lost, the second-best class is expected to ultimately suffer the same fate, and so on. Felsenstein (1974) called this phenomenon Muller's ratchet". We have now clarified that we are discussing Muller's ratchet in this section.

Later in the discussion, we then discuss ROH (runs of homozygosity) which likely arose from gene conversion/mitotic recombination. Gene conversion can be a method of asexual organisms ridding themselves of deleterious mutations. Here, as mentioned by the reviewer, selection will ensure a fitter allele on one haplotype will be selected for in the next generation. HOWEVER, following the introduction of ROH the population will have reduced long-term fitness as the homozygosity will contain some sub-optimal alleles, and the genetic diversity which would have been present in the population through heterozygosity at these sites is now absent from the population. This is thought to have an even greater effect than the accumulation of deleterious mutations and so will drive Muller's ratchet even quicker than de novo mutations.

It should be noted that for trypanosomes, all of this has been discussed previously for the asexual *T.b. gambiense* Group 1 (<https://elifesciences.org/articles/11473#s3>).

In summary, therefore, we do not contradict ourselves in the discussion, but we appreciate the opportunity to clarify ourselves

Odd organisation as we are suddenly given data with no information on how obtained, in what I assume is the introduction?

This was a result of the direct journal transfer process. We have now formatted the paper according to the Nature Communications journal guidelines.

Depending on the depth of sequencing SNPs can be hugely overestimated, especially with short reads. How has this been controlled?

Assuming this comment relates to the SNP density as part of Fig 1b, we agree that the depth of reads can have a large impact on SNP overestimation. However, in the SNP track we used an equal number of SNPs for each lineage. These were randomly selected and give an indication of the density for each lineage. This has been commented on in the methods section (line 675-678). We have also included read counts for each isolate in Supplementary data file S1.

I'm a bit unclear as well as to why the phylogeny is based on SNPs and not full genome data? Is this simply a diversity phylogeny (not sure what the correct term would be here) rather than a true one?

The SNPs were called from full genome data. This is the standard method for generating a phylogenetic tree for this kind of data. An alternative approach would be to assemble the genome of each isolate, annotate the genes, extract each protein sequence, identify single copy orthologues concatenate these orthologous sequences for each isolate and then feed these concatenated sequences into the same tool we used for generating our tree. This alternative approach would have major downsides in terms of computational time to assemble each isolate and annotate it, significant errors and mis-assemblies in the genome assemblies and missing annotations etc. The tool used to build the phylogenetic tree would then need to remove all the non-variable sites in the concatenated sequence (of which there would be a huge number as these isolates are closely related), leaving polymorphic sites between the isolates – otherwise known as SNPs – which you would then use to build the phylogenetic tree.

Instead of this torturous process, the standard method for building phylogenetic trees is to identify variants against a reference sequence and use these variant sites to build the tree. It is important to note that the nucleotide will be recorded for each isolate in the analysis, even if only one isolate has a SNP at that site.

Furthermore, our methods are consistent with previous analysis of the phylogeny of *T. brucei* using genome data where SNP based trees were also used. Hence our approach is the most relevant and accepted for this kind of analysis.

BUT, do we not already know that these strains have distinct clade structure? How does this new analysis compare?

We do indeed! Although we were one of the most recent authors to analyse the clade structure of trypanosome lineages using genome data (Oldrieve, et al, *Microb Genom* (2021) PMID 3439734) we were by no means the first (e.g. Carnes et al, *PLOS NTD* (2015), PMID 25568942; Cuypers et al, *Genome Biol Evol* (2017) PMID 28541535). As we noted 133-134, our analyses corroborate these previous studies in terms of the polyphyletic origins for monomorphic lineages. But it was important to establish this phylogeny with all of the isolates used in our study so that clades could be defined and considered individually and therefore relevant SNPs potentially contributing to loss of pleomorphism identified.

It may be my ignorance but I am not clear that you can equate nonsynonymous mutations exclusively with deleterious mutations.

It is correct that nonsynonymous mutations are not exclusively deleterious, and they may be neutral or, sometimes, advantageous. The context of the referee's comment was not completely clear, but the ratio of synonymous to non-synonymous mutations (a measure we used to narrow down potentially important changes in monomorphic lineages) does give a measure of evolutionary selection and our analyses used this to identify such changes in either direction. We did not equate non-synonymous mutations exclusively with deleterious changes.

I'm also unclear if the binning of data with vastly different numbers of strains (which I assume is the case) can be a major impact here, i.e. one clade has pneumonia strain, another has over ten.

As the reviewer has not included any context to their comment, it is hard to address.

Nonetheless, we agree broadly that we have used different numbers of isolates for each clade (because we used all those available). We have normalised the presentation of data as much as possible e.g. plotting the density of SNPs and dN/dS values rather than absolute numbers.

However, ultimately it is unavoidable to compare between different groups with different representation because of the rarity of some clades e.g. *T. b. evansi* type B. Extensive and exhaustive field sampling would be required to generate greater representation of rare clades but this is clearly beyond the scope of this study. Nonetheless, we present the most complete monomorphic dataset produced to date and have analysed the data and strains available to us.

There is a major hotspot of ChrII and others - what do these represent?

Several chromosomes show regions enriched for SNPs, runs of heterozygosity or copy number variation. We have not explored all these genes and regions, instead using precisely defined criteria to focus on molecules that could contribute to life cycle simplification- a strategy that our experimental validations show has been successful. Bespoke analyses of different biological processes can be carried out by readers because we have made the full analysis and datasets publicly available.

Is it not expected that mitochondrial components will be significantly affected, but if these are neutral and have occurred independently that these would differ?

Mitochondrial components are significantly affected as shown by the GO term analysis in Supplementary figure 1. Further, distinct lineages show distinct GO term enrichments, consistent with the independent origins of each.

There is no real statistical analysis, for SNPs, genome features or the prevalence of GO terms.

GO terms – only significantly enriched GO terms are reported (Benjamini 0.05 cutoff). We have added this to the methods for clarity.

Genome features and SNPs – as the reviewer commented on previously, we have an unequal number of isolates for each clade and the low number of isolates in the *T.b. evansi* type B and IVM-t1 clade prevent statistical comparison. However, we filtered our data to highlight genes with clade specific mutations and validated our survey of these genomic data experimentally. As stated above, we present the most complete monomorphic dataset produced to date and have analysed all the data and strains available to us.

ROH are cited as a possible feature to purge deleterious mutations, in contradiction to the statement in the introduction.

We have addressed the comment regarding deleterious mutations above, highlighting that there is no contradiction between the Introduction and Discussion.

with cell cycle development and expression of the stumpy-specific marker protein PAD1
PAD1 can be upregulated by stress, and I am unclear as to what 'cell cycle development' means.

It has been argued that PAD1 expression can be activated by stress. However, PAD1 expression is the only accepted molecular marker for the stumpy form described and so is an important and relevant assay when examining developmental competence in the parasites. Moreover, recent analysis has demonstrated that the pathways of stumpy formation and stress response are quite different (Cayla et al, Nature Communications 2024, PMID 38582942). This point is also made by the fact that re-expression of key RNA regulators in selected monomorphs generates cell cycle arrest – i.e. early steps in the differentiation pathway- but not PAD1 expression.

'Cell cycle development' is progression through a proliferative cell cycle. We apologise if our phraseology was clumsy and have corrected it.

Need to be clearer as to what the changes made to the selected genes are. Usual to call these out as, eg, G259Y or similar rather than 'clade' which I cannot see as accurate.

The mutations detailed are clade specific and hence it is appropriate to discuss both the mutation and the clade. For example, in the allelic replacement experiments, all the changes in the *T. b. evansi* type A Tb927.2.4020 allele were introduced (these detailed in the supplementary figure 4a), and these were present in all members of that clade, but distinct from those in *T. brucei* 927/4 or the *T. b. equiperdum* BoTat clade. Hence it was appropriate to describe this as the *T. b. evansi* type A clade specific allele.

Also, helpful if the accessions were accompanied by the annotation for that gene. Do the mutations lead to destabilisation, truncation or?

In supplementary Table S2 and associated legend we detailed the full annotation of category 1 and category 2 genes, with respect to their functional descriptors and the nature of the mutations detected in the respective lineages:

- Category 1 (Tab 1) Genes which have a clade specific high-impact or moderate-impact mutation in a monomorphic clade in a known QS pathway gene. The gene must also display no high impact mutations in any pleomorphic isolates.
- Category 2 (Tab 2). Genes which have a clade specific high-impact or moderate-impact mutation in a monomorphic clade along with a dN/dS ratio <1 in pleomorphic isolates and >1 in any monomorphic clade. The gene must also have a smaller log fold change in D3, D6 and PF than in the DIF category whilst displaying a log fold change in the DIF category greater than -1.5. The gene must also display no high impact mutations in any pleomorphic isolates.

In the absence of specific reagents to detect the proteins of each mutated gene we cannot assess stability data; however alpha fold analysis of the candidate genes that were experimentally evaluated revealed no significant predicted structural changes (shown in Supplementary Figure s4)

All three seem to have been implicated in differentiation by an earlier high throughput screen. This should be mentioned; it adds some weight to the arguments here.

Indeed, it was reassuring that genes that we had identified as important in QS previously were detected as mutated in monomorphic lineages. However, of those experimentally evaluated, only Tb927.2.4020 had a previously known role in quorum sensing/stumpy formation. The exciting outcome of our study was that previously unidentified contributors to developmental regulation (and other aspects of the monomorphic lifestyle, such as altered motility) could be identified in our genomic analysis of monomorphic lineages. One of these was Tb927.5.2580 – identified as a completely new and unexpected developmental regulator. We expect our datasets will provide a rich seam of further developmental regulators that were previously unknown- these will emerge using long term experimental analysis similar to that we have employed for the exemplar genes evaluated in this study.

Given that the three genes analysed are involved in vastly distinct processes, does seem a need for a unifying concept here., Also, as 'brucei' is odd that there are such differing responses.

Indeed, different processes are modified in the evolution of monomorphism in these distinct field isolates. This is not surprising given that the monomorphic clades arose independently and have different histories and evolutionary trajectories at the point of analysis. It was for this reason that the changes identified do not highlight a common set of changes in all the lineages, which is one of the findings from our study. To explore how monomorphism could originally arise was exactly the reason for our complementary analysis of the de novo laboratory selected lineages, which were independently generated from the same origin, selection strategy and timescale (unlike the field isolates). This revealed that the expression of a small number of key RNA regulators were consistently perturbed in every lineage, and by re-expression we could demonstrate the importance of these in stumpy formation. Hence the unifying concept (as detailed in Figure 5) is that initially reversible gene expression changes provide the flexibility for non-tsetse transmission (proto monomorphs) before lineages become locked-into the monomorphic lifestyle through mutations that can arise independently in distinct clades in different processes important in stumpy formation. This was represented in Figure 5.

With respect to *T. brucei* being 'odd', it is certainly the case that *T. brucei* but not *T. congolense* has adapted to tsetse independent transmission. However, *T. vivax* also shows flexibility in its use of either tsetse dependent or independent transmission.

The lethal impact of BHI on all of the 927.11.3400 is spectacular, but how reconcile with life cycle progression in *T. brucei* itself?

In fact, BHI is not lethal- it simply rapidly arrests the parasites through their detection of the quorum sensing signal. Hence, the data in Figure 2e (a cumulative growth curve) reflects arrest as they develop to stumpy forms.

Why only Tb927.5.2580 analysed in more detail?

Tb927.5.2580 was analysed in vivo since this was a novel regulator of life-cycle development identified in our screen and so it was important to validate its developmental defect in response to the physiological quorum sensing signal in vivo, rather than simply BHI in vitro. Tb927.2.2040 was not reanalysed because it has previously been shown to affect stumpy formation in vivo. Moreover, this line showed very low infectivity in mice- something that can occur after long term passage/transfection of trypanosomes. Tb927.11.3400 was shown in vitro not to have a developmental defect and so was not analysed in vivo for stumpy formation.

Reviewer #2 (Remarks to the Author):

The group of trypanosomes *T. brucei sensu lato* has a number of peculiar features and one of them is the ability of some strains to eliminate the infamous tse-tse fly vector from their life cycle, with major consequences for their spreading outside the so-called tse-tse belt. While this phenomenon was recently studied from several perspectives, the authors of this paper took a new one, which this reviewer appreciates.

They focused on mutations potentially associated with the loss of stumpy formation, changes in quorum sensing and PAD1 expression, as factors that may play critical role in the switch from polymorphism to monomorphism (the involvement of flagellar mobility in this context is another novelty, at least for this reviewer). Using a set of *T. brucei equiperdum* and *T. b. evansi* strains (probably the most comprehensive available) as well as experimental reversion of *T. b. brucei* to monomorphism, and - importantly - expression of the potentially most interesting genes from the dys/akinetoplastic trypanosomes in *T. b. brucei*, and the rescue thereof. Combination of these approaches allowed them to identify mutations in known and novel components of quorum sensing, and other genes worth follow-up studies.

The results are not black & white, as the individual identified mutations have either different importance in different strains/ecotypes, or work in tandem with other identified mutations or some yet unknown ones (some of the add-backs worked "half way", but that's not unusual for add-backs in *T. brucei*).

Indeed, the 'half-way' response for add-back phenotypes is common for trypanosome transfection experiments. However, when analysing developmental phenotypes this is particularly acute because selection after multiple rounds of transfection and *in vitro* passage can rapidly reduce the level of stumpy formation. This is why we were particularly careful in our experimental strategy to always transfect, at each step, pleomorphic parasites in parallel, replacing wild type alleles with wild type alleles, to control for culture-associated loss of pleomorphism and potential disruption of gene function, which is essential for CRISPR manipulations.

Moreover, there are apparently various paths to monomorphism, which overlap only to some extent. But overall the progress achieved by this study qualifies it, in the eyes of this reviewer, for Nat. Commun.

This is exactly our conclusion, and thank you for your enthusiasm for our study.

I am pretty happy with the experimental setup which is elegant, and presentation of the data, but I have a number of suggestions to be considered (especially for the introduction and discussion), as well as some omissions to be fixed.

58 - They start with the subspecies status, but it was shown recently that in the case of *T. brucei*, these are not subspecies. Even if they have hard time to accept that, they should at least mention that another alternative (a way better one in the view of this reviewer) has been published in Trends in Parasitology recently, namely that of the ecotypes.

As also highlighted in response to referee 1, indeed, the nomenclature surrounding pleomorphic and monomorphic trypanosomes has been controversial, with several solutions having been proposed to resolve the complexity of differing phenotypes, genetic exchangeability, disease presentation and host specificity, transmission modes and the polyphyletic origins of monomorphic lineages. Among the most recent is the proposal of ecotypes. This is not yet universally accepted in the field as the favoured approach to resolve the issue, and there are complications when laboratory selection is used to alter characteristics that define ecotypes – as we have done when selecting monomorphic lines from pleomorphic lines in BHI where, for example, changes may be reversible/epigenetically controlled. Instead, we consider it is valuable to acknowledge the genetic relationships and phylogeny of different parasite lineages and so refer to subspecies and 'types' in our manuscript (which is also not without issues). Nonetheless, the introduction of ecotypes is a valuable proposal, and we are happy to acknowledge and cite the relevant publication, which also usefully debates the challenges with nomenclature for these parasites. Ultimately, the field as a whole will be required to find a consensus.

71 - be concise - lifecycle versus life cycle (73), geographic vs geographical, nonsynonymous vs non-synonymous etc.

Changed- thank you

74 - a reference to the Lun paper (Trends in Parasitology) would be appropriate here

Thank you- now included.

78 - biting flies sounds a bit inaccurate, tabanid flies would be better but if they are aware of any non-tabanids involved, then it is fine as is

There is evidence that *Stomoxys* (biting stable flies) are capable of transmission as well as tabanids. Example references are; PMID: 30730048; PMID: 25229294; PMID: 9824826; PMID: 8525279; PMID: 7831761. We have clarified this in the text however, since we agree 'biting flies' is less accurate than 'blood feeding biting flies'.

93 - again - types, subspecies, ecotypes...

Thank you. As detailed above, this alternative nomenclature is now cited.

105 - reference to the review paper in TiP 2022, where this was shown very convincingly would be in place here

Thank you. As detailed above, this alternative nomenclature is now cited.

140 - I missed in the discussion any comments/elaboration on the accumulation of mutations in the mitoproteome

We highlight in the text the GO enrichment of changes in the nuclear encoded mitochondrial genes (also detailed in Supplementary Figure 1) but now also refer to the (expected) prevalence of such changes in the discussion. Our specific intention was to identify changes that contribute to life cycle simplification- namely reduced stumpy formation- and we have focused on that area. But our data will provide a rich seam of information which can be mined by many researchers to focus on their own specific interests relating to the adaptations that monomorphic parasites adopt to favour their alternative life style (transmission, host/niche and disease presentation) or where there is a degradation in competence for processes only required in the fly.

151-153 - any idea why these GO terms were overrepresented?

GO terms are associated with the presence of the three TDPXs (tryparedoxin-dependent peroxidases) which are part of the antioxidant defence system and localise to the mitochondrion and cytosol. We have now included this in the manuscript. We do not have further data on this, which could be a useful area for us or others to explore in future.

216 - perhaps appropriate mentioning here that complex IV subunits are present, albeit at low level, in bloodstreams (Zikova et al., PLoS Pathog. 2017).

Although the proteomic detection of complex IV subunits is variable in different studies, we have modified our text as suggested and cited the relevant study.

235 - a high throughput – typo
Changed- thank you

335 - Muller's ratchet is correct
Changed- thank you

336-339 - it puzzles me why for the support of their view of gradual switch to monomorphism they do not cite paper(s) that show exactly that on the level of kDNA?!

Monomorphism arises independently of the gradual loss of kDNA; these are independent events and processes. Our data, and that of others, demonstrates monomorphs can arise without kDNA loss and kDNA loss can occur without monomorphism (Dewar, C. E. et al. Mitochondrial DNA is critical for longevity and metabolism of transmission stage *Trypanosoma brucei*. PLoS Pathog 14, e1007195 (2018). <https://doi.org/10.1371/journal.ppat.1007195>). Nonetheless, we are happy to cite the progressive, though independent nature, of each process (Lai DH, Hashimi H, Lun ZR, Ayala FJ, Lukes J. Adaptations of *Trypanosoma brucei* to gradual loss of kinetoplast DNA: *Trypanosoma equiperdum* and *Trypanosoma evansi* are petite mutants of *T. brucei*. Proc Natl Acad Sci U S A. 2008 Feb 12;105(6):1999-2004. doi: 10.1073/pnas.0711799105. Epub 2008 Feb 1. PMID: 18245376; PMCID: PMC2538871).

342-344 - they may strengthen their interpretation of their data by including reference to a paper from Zhao-Rong Lun and collaborators, who already predicted the continuous emergence of new monomorphic strains out of Africa.

Although we consider that that study had important limitations because of the nature of the analyses and lineages analysed, we agree it is valuable and appropriate to reference the study, particularly in relation to the emergent threat of non-tsetse transmitted human infective trypanosomes. A reference to the paper is now included in the revised manuscript.

some small issues with references - double check Hoare 1970, missing number of the paper for 24, errors in 27, 29 and probably others

Hoare is 1972, not 1970, but nonetheless we have now updated to include the author initials and publisher details.

The DOI is the appropriate reference for Reference 24 but as requested we have inserted the page numbering (1-8)

Other references highlighted have been checked and amended, though EndNote tends to 'recorrect' our changes. We will ensure referencing is fully correct at proofing stage upon acceptance.